# A Contention-Free Cooperative MAC Protocol for Eliminating Heterogenous Collisions in Vehicular Ad Hoc Networks

**DOI:** 10.3390/s23021033

**Published:** 2023-01-16

**Authors:** Nyi Nyi Linn, Kai Liu, Qiang Gao

**Affiliations:** 1School of Electronics and Information Engineering, Beihang University, Beijing 100191, China; 2Hangzhou Innovation Institute, Beihang University, Hangzhou 310051, China

**Keywords:** vehicular ad hoc networks, medium access control, cooperative forwarding, vectorized trajectory estimation, heterogenous collision resolution, access delay

## Abstract

In vehicular ad hoc networks (VANETs), efficient data dissemination to a specified number of vehicles with minimum collisions and limited access delay is critical for accident prevention in road safety. However, packet collisions have a significant impact on access delay, and they may lead to unanticipated link failure when a range of diversified collisions are combined due to complex traffic conditions and rapid changes in network topology. In this paper, we propose a distributed contention-free cooperative medium access control (CFC-MAC) protocol to reduce heterogenous collisions and unintended access delay in stochastic traffic scenarios. Firstly, we develop a cooperative communication system model and cooperative forwarding mechanism to explore the optimum road path between the source and destination by identifying the potential cooperative vehicles. Secondly, we propose a vectorized trajectory estimation mechanism to suppress merging collisions by identifying the relative velocity of vehicles with different speeds in a specific time interval. Based on the case study, typical heterogeneous collisions and aggregated heterogeneous collisions at dissociated positions and associated positions are investigated. In both cases, we propose the corresponding collision-resolving mechanisms by methodically recapturing the colliding time slot or acquiring the available free time slots after identifying the access vehicles and comparing the received signal strengths. Performance analysis for collision probability and access delay is conducted. Finally, the simulation results show that the proposed protocol can achieve deterministic access delay and a minimal collision rate, substantially outperforming the existing solutions.

## 1. Introduction

In recent years, the intelligent transportation system (ITS) has been developed to alleviate the number of road fatalities, accidents, and traffic jams [1,2]. Vehicular ad hoc networks (VANETs) are emerging as a new landscape of mobile ad hoc networks (MANETs); they aim to provide a wide spectrum of safety and comfort applications to drivers and passengers, and are regarded as an essential component in developing smart cities and applications, ranging from safety to entertainment services in ITS [3,4,5].

For the purpose of seamlessly connecting vehicles, IEEE 802.11p-based wireless communication technologies have been developed, called dedicated short-range communications (DSRC) in the US and ITS-G5 in the European Union (EU) [6,7]. As one option, the 3rd Generation Partnership Project (3GPP) has completed the first version of the Long-Term Evolution Vehicle-to-Everything (LTE-V2X) technology in the 4G-LTE network and has been investigating new use cases and technology requirements in 5G, 6G, and 6G-and-beyond networks [8,9].

DSRC/ITS-G5 or LTE/5G V2X technologies bring forth vehicle-to-infrastructure (V2I), infrastructure-to-vehicle (I2V), vehicle-to-vehicle (V2V), and V2X communications, which are predominant building blocks for future ITS applications [4,5,6,7,8,9,10]. Observing the specific dynamics of vehicular mobility, along with the local scope targeted by DSRC, ITS-G5, LTE/5G V2X, and the AI-empowered intelligent reflecting surface (IRS) communication framework [11] for ITS applications, notable protocol stacks have been developed by IEEE Wireless Access in Vehicular Environments (WAVE), ISO Communications Access for Land Mobiles (CALM), and the European Telecommunications Standards Institute (ETSI) [5,12].

In the VANET environment, medium access control (MAC) protocols play a critical role in providing reliable communications and disseminating important safety messages (SMs) to avoid chain reactions and catastrophes by efficiently utilizing the shared wireless channel. Providing an efficient MAC protocol is a challenge because of the high speed of the vehicles, frequent topology changes, insufficient infrastructure, and wide range of quality-of-service (QoS) requirements [13,14,15,16]. Time division multiple access (TDMA)-based MAC protocols are becoming an emerging research topic in the area of VANETs, in which time is divided into frames and each frame is divided into time slots, and various vehicles can access different time slots [17,18].

Obviously, TDMA-based MAC protocols for VANETs are categorized as contention-based and contention-free strategies [16]. Since contention-based IEEE802.11p has been developed, DSRC has adopted carrier-sense multiple access with collision avoidance (CSMA/CA) mechanisms to avoid collisions by beginning transmission only after the channel is sensed to be idle [5,6]. However, CSMA/CA is affected by serious collisions in cases of heavy traffic load, and it is not effective in providing shared high-data-rate transmissions [9,19]. Furthermore, IEEE 802.11p has been standardized with enhanced distributed channel access (EDCA) to handle the problem of rapid network topology and vehicle mobility [20,21]. However, the IEEE 802.11p protocol suffers from poor scalability and involves the problems of hidden and exposed terminals, which results in the failure of real-time message delivery [6].

The contention-based protocols do not have pre-scheduled transmissions. As a matter of fact, when a vehicle wishes to send its data packets, it competes for channel access without guaranteeing successful packet transmissions. They also have some limitations that cause broadcast storm problems, severe packet collisions, large average coverage delay, and diminished reachability [20,22]. In contrast, the main idea of contention-free schemes is to eliminate unnecessary collisions and avoid predicaments via the periodic exchange of control messages to maintain the scheduled table and network time in synchronization [17]. Therefore, we focus on the contention-free approach to provide short latency, a significant decrease in packet loss, and deterministic access delay in real-time applications.

It is undeniable that delay is one of the most crucial metrics to evaluate the QoS of a network [22,23]. In principle, the delay for successfully transmitting a data packet from one vehicle to another consists of various elements, including processing delay, queuing delay, local delay, and access delay over a link [23]. In general, processing delay and queuing delay are negligible since they are in the order of tens of milliseconds and are relatively smaller than other delay factors [24]. Local delay can also be omitted since it is mostly concerned with contention-based failed channel transmission quality [25]. However, access delay has a significant impact on packet collisions because it is the average time from the moment when a vehicle starts attempting to send a packet until the beginning of its successful packet transmission [20,23]. Therefore, in this paper, we focus on diminishing the access delay to prevent unexpected link failure when a variety of distinctive collisions are aggregated interchangeably.

Although many existing studies [24,26,27,28,29,30,31,32,33,34,35] have been devoted to mitigating certain kinds of collisions in VANET, there are still two major issues that have not been completely addressed. First, the majority of these studies focus on distributed vehicle communication mechanisms, but there has not yet been any extensive research on the thematic analysis of heterogenous collisions to lessen cumulative access delay since safety applications have stringent QoS requirements in VANETs. Second, the deployment of existing slot-partitioning techniques remains a challenge in having deterministic access delay in highly dynamic and stochastic traffic flow, and may lead to slot redundancy and link failure. In this paper, we propose a novel distributed contention-free cooperative MAC (CFC-MAC) protocol; moreover, we systemically investigate the suppression of typical heterogenous collisions and aggregated heterogenous collisions in hidden terminals and exposed terminals and diminish the merging collisions of cooperative vehicles, which may offer deterministic access delay with minimal collisions to mitigate unnecessary retransmissions. The contributions of this paper are as follows:We develop an optimized traffic model to forward data packets by effectively utilizing cooperative vehicles to improve the probability of successful packet transmission. A limited number of optimum cooperative vehicles are carefully selected by determining the closest position to the midpoint between the source vehicle and the destination vehicle.To reduce merging collisions to adapt to the frequent converging and diverging of rapid topology changes in the network due to high vehicle mobility, we propose a vectorized trajectory estimation (VTE) method by comparing the relative velocity of vehicles within a specific time interval by identifying the previous positions.Cooperation collisions occur when the cooperative vehicle and relay vehicle are contending to reserve the same time slot. Based on the distinct stochastic traffic scenarios of the proposed system model, we investigate two cases of heterogenous collisions: (1) typical heterogeneous collisions, such as cooperation collisions, access collisions, relay collisions, and reservation collisions; (2) aggregated heterogeneous collisions of the hidden terminals and exposed terminals at dissociated positions (occurring simultaneously but quite independently) and associated positions, respectively. In both cases, we propose the relevant collision-resolving mechanisms of time slot assignment procedures by methodically operating the four distinctive packets after identifying the access vehicles and comparing the received signal strengths.We validate the protocol performance by analyzing the collision probability, packet reception probability, and access delay of diversified heterogenous collisions in stochastic traffic scenarios. Extensive simulation experiments are conducted under various traffic conditions with different vehicular networking settings.

The rest of the paper is organized as follows. Section 2 presents an overview of the related work. Section 3 describes the basic concepts and assumptions and the proposed system model in depth. In Section 4, we provide the details of the proposed CFC-MAC protocol concerning the techniques for time slot assignment procedures, packet operations, and collisions resolution, as well as the time slot reallocation mechanisms for reducing typical heterogenous collisions and aggregated heterogeneous collisions of hidden and exposed terminals, and merging collisions. In Section 5, performance analysis in terms of collision probability, packet reception probability and access delay is described. Section 6 presents the simulation results and a performance comparison. Section 7 briefly discusses the significant points of our work. In Section 8, we draw conclusions and present ideas for future work.

## 2. Related Work

Based on cooperative communication mechanisms and distinctive resource allocation procedures, several TDMA-based MAC protocols have been proposed recently to resolve various types of packet collision problems, link failures, and the reusability of time slot allocation. Two of the most promising classifications developed for TDMA-based contention-free MAC protocols are centralized and distributed schemes [18,30]. In centralized TDMA-based contention-free MAC protocols, the central coordinators, i.e., roadside units (RSUs) or cluster heads (CHs), allocate and control message forwarding in the network [22,31,36]. In the RSU-based approach, RSUs act as intermediate relays for inter-cluster communication. Since a large number of RSUs are required to facilitate high traffic, this approach is not suitable for highway roads [37]. For the cluster-based approach, the clusters have to form and change over time in high-mobility scenarios, and the dramatic change in vehicle mobility has an impact on cluster stability due to cluster splitting and merging, which results in consequential network overheads and performance degradation [22,36,38].

In distributed TDMA-based contention-free MAC protocols, the cooperative communication-based method is an excellent way to improve transmission performance in high-density traffic as compared to low-density traffic [17,35]. Based on this, [20,22,24,39,40] propose increasing channel capacity in VANETs and satisfying the latency requirements of various applications. In multi-hop wireless networks, cooperative communication is an effective and successful way to improve performance [20,41,42]. Therefore, we adopt the contention-free cooperative approach since these forms of strategy are highly scalable and fault-tolerant. In survey works [4,10,17,42], the selection of cooperative vehicles is the predominant factor in transmission efficiency, and the vehicle with the highest transmission rate will be selected as the third-party cooperative vehicle. Generally, the heterogeneity of collisions and delay demands is unexplored.

In contention-free cooperation-based MAC protocols, it is common for vehicles to communicate with each other on the basis of one-hop distance, and the time slot is randomly reserved for each vehicle in the network. Afterward, the cooperating vehicles or RSUs forward the messages that are multi-hop toward the destination vehicle V_D_ from the source vehicle V_S_ [17,27,34,40,41,42]. EVC-TDMA MAC [28] selects the ideal forwarding vehicle in multi-hop message transmission by dynamically selecting the relay vehicles based on the relative speeds and buffer sizes, and resolves the dramatic merging collisions. Its drawback is that the packet dropping rate increases in highly dense traffic by removing the header packets when the buffer is full. In addition, it may not resolve the hidden and exposed terminal problems. In NC-MAC [33], the preamble-based feedback mechanism, retransmissions, and network coding are combined to enhance broadcasting reliability to support V2V beacon broadcasting. However, the protocol design only supports one-hop transmissions with the increased probability of linear combinations of undesirable beacons, which may incur performance loss in highly dense scenarios.

In ECTOB-MAC [43], the expected broadcast performance is enhanced by designating cooperators with the maximum expected single-hop broadcast efficiency (SBE) value for a single relay. However, redundant transmissions from multiple relays may not be avoided since the expected waiting time for a relay candidate can be interfered with by some unexpected transmissions from a hidden terminal of the previous forwarder. TMR-MAC [44] can identify an optimized path with a short round-trip time (RTT) threshold for packet delivery to V_D_ with a minimal time delay. When one path fails, data packets are sent through the next least RTT path. However, an increasing delay occurs when RSU sends the packet using the first minimum RTT path in its routing table and waits for the acknowledgment from the V_D_. MCDP-MAC [45] relies on the forward-if-relevant principle to dynamically determine the zone of relevance (ZOR) of the event. Its major disadvantage is that there is no stringent mechanism whenever a new event arrives to resolve the access collisions and broadcast storms.

In DARP MAC [30], each vehicle coordinates its channel access with its neighbors to address the hidden terminal problem. Different preamble types are used to facilitate distributed reservation, identify beacon collisions, and resolve collisions to guarantee the dependability of beacon broadcasting. Its disadvantage is that it is only concerned with the hidden terminals for single-hop beacon broadcasting, and does not consider exposed terminals and merging collisions. In PeerProbe-MAC [46], the online neighbor distribution estimate approach is proposed, and vehicles collaborate to concurrently probe their neighborhood via wireless symbol-level communication. Although it is scalable and capable of correctly reconstructing highly dynamic neighbor distributions under harsh channel conditions, the signal-to-noise ratio (SNR)-based adaptive de-mapping strategy is insufficient to decrease the bit error rate (BER) in the most catastrophic channel conditions.

A comparison of the most recent studies related to our work is shown in Table 1. For instance, the studies conducted in [24,33,43] partially cover resolving hidden terminals and access collisions, but others do not discuss exposed terminals and different types of collisions in the field of cooperative approaches. Obviously, the work in [28] only considers resolving merging collisions, whereas [45,46] does not resolve any of the collisions.

Despite the fact that the aforementioned contention-free cooperative MAC protocols are conducted to support transmission performance in multi-hop networks, the majority of these protocols are still unable to comply with the highly dynamic topology and frequently disconnected network links. Moreover, none of them figure out the typical heterogeneous collisions, such as cooperation collisions, access collisions, relay collisions and reservation collisions, and also do not consider the cumulative access failure of network scenarios in the aggregated heterogeneous collisions of hidden terminals, exposed terminals, and merging collisions. Motivated by these issues, we design CFC-MAC in this paper by reducing the typical heterogenous collisions and aggregated heterogenous collisions to avoid chain collisions and guarantee optimal access delay.

## 3. System Model

This section describes the fundamental assumptions and the proposed cooperative communication system model with a typical traffic scenario.

### 3.1. Fundamental Assumptions

In our CFC-MAC, similar to 5G C-V2X [33], it is assumed that the network is distributed and all vehicles have equivalent capacities. Vehicles utilize the same transmission power to communicate with each other via DSRC/WAVE operation mode in the range of 5.850~5.925 GHz, which can actively regulate transmission power to alter the communication range, provided that there is one control channel (CCH) for system control and SMs, and six service channels (SCHs) for the exchange of non-safety messages (NSMs) [21]. The proposed system model is composed under the following assumptions:Communication: To provide reliable communications, we consider that each vehicle is equipped with two transceivers: one transceiver for CCH to listen for the channel status and SM transmission, and the other transceiver for SCH to send NSMs of infotainment services. Our proposed model primarily outlines how to access CCH to broadcast SMs. As depicted in Figure 1, the channel time is divided into synchronization intervals (SIs) of the TDMA frames with a fixed length of 100 ms. Each TDMA frame is split into two sets of time slots: the left CCH interval (L-CCHI) and right CCH interval (R-CCHI); these correspond to the vehicles going in the left and right directions, respectively. A guard interval of 4 ms is appended at the end of each interval set for synchronization.Synchronization: Each vehicle is equipped with a global positioning system (GPS) receiver to provide a synchronous reference time. Because low-end GPS receivers can provide a one-pulse-per-second (PPS) signal with an accuracy of fewer than 100 ns and the GPS oscillator is stable, vehicles may preserve synchronization for a while even if the GPS signal is lost temporarily.Computation: For road safety, the minimum headway between all vehicles on the road must follow the 2s rule, which means that the distance for V2V connectivity must be equal to 2 times the vehicle velocity [47,48,49]. Vehicles are capable of performing fundamental operations, such as generating random numbers, hash functions (by using this, hashing algorithms can convert data with arbitrary length to a fixed length), and simple matrix algebra calculations.

### 3.2. System Model of Cooperative Relay

In VANETs, it is a common phenomenon that a vehicle wants to exchange messages with other vehicles out of its communication radius *R*_c_, which is called multi-hop communication. We first propose the basic idea of the cooperative relay for multi-hop communication in an ideal traffic scenario, as shown in Figure 2. When *R*_c_ is the same for each vehicle, the distance between the source vehicle V_S_ and the destination vehicle V_D_ is defined as *D*_SD_, and the distance of the potential cooperative vehicle from V_S_ is defined as *D*_C_. It can be clearly seen that V_S_ is unable to transfer data directly to V_D_ due to the limitation of the transmission range. Depending on the *R*_c_ between each of the vehicles, all vehicles can be classified into five categories: the source vehicle V_S_, the destination vehicle V_D_, the relay vehicle V_R_, the cooperative vehicle V_C,_ and the optimum cooperative vehicle V_OC_.

V_S_ can transmit data packets generated by itself, whereas V_D_ is not in its *R*_c_, and the relay vehicle V_R_ can deliver multi-hop relay data packets generated by other vehicles whether it is in the range of *D*_SD_ or not. A relay vehicle V_R_ can be transformed into a cooperative vehicle V_C_ when it belongs to the transmission range of *D*_SD_. Very few numbers of cooperative vehicles V_C_ are prone to becoming the optimum cooperative vehicle V_OC_ when they are effectively selected by determining the closest position to the midpoint within the minimum distance of *D*_SD_. A detailed selection procedure for V_OC_ will be discussed in the next section.

In fact, the channel characteristics for V2V and V2I communication must be determined using propagation models. For the free-space propagation of an electromagnetic wave between two vehicles, we consider the wireless channel model based on the Friis transmission formula [50]. The received signal power $P_{r}$ of a vehicle can be given by
(1)$$P_{r}=P_{t}\frac{G_{r}G_{t}}{L}{\left(\frac{\lambda }{4\pi^{2}}\right)}^{2}{\left(\frac{1}{D_{SD}}\right)}^{\gamma },$$
where $P_{t}$ is the transmitted power, and $G_{t}$ and $G_{r}$ represent the transmitter and receiver gains, respectively. λ is the signal wavelength, *γ* denotes the path loss exponent, and *L* is the system attenuation loss.

Assuming that each vehicle antenna is isotropic and has no directivity, the free-space path loss *ℓ* between the transmitter and receiver can be defined as the loss factor in the ratio of the transmitted power to the received power as $\ell (x_{tx},x_{rx})=P_{t}/P_{r}={\left(4\pi D_{SD}/\lambda \right)}^{2}$. We can express the signal-to-interference-plus-noise ratio (SINR) at the vehicle receiver as
(2)$$\mathrm{SIN}R=\frac{P_{t}K_{0}\ell (x_{tx},x_{rx})}{\sum K_{x}\ell (x_{i},x_{rx})+N},$$
where $\ell (x_{i},x_{rx})$ is the path loss between the interferer and receiver, and *N* is the noise power of additive white Gaussian noise. *K*_0_ denotes a constant that depends on the channel and antenna characteristics, and *K*_x_ represents a constant that depends on the interfering channel fading for an interferer at location *x*. If the SINR is greater than the preset threshold *β*, i.e., SINR ≥ *β*, the received vehicle is regarded as a productive neighbor, and a packet is expected to be successfully received.

## 4. CFC-MAC Protocol

In this section, to visualize the proposed CFC-MAC protocol design, a structured flowchart is shown in Figure 3. First, we propose the cooperative forwarding mechanism (CFM) to determine the optimum cooperative vehicle to minimize excessive retransmissions for improving transmission efficiency. Next, we define four types of packet based on the different preambles to discover the status of the specific vehicle and perform the relevant execution. Actually, they are employed to support the resolving of the diversified heterogenous collisions. 

Then, we scrutinize the merging collisions of the proposed cooperative system model and provide a vectorized trajectory estimation (VTE) mechanism by identifying the relative velocity of vehicles with different speeds in a specific time interval. Moreover, we thematically analyze the typical heterogeneous collisions and aggregated heterogeneous collisions at dissociated positions and associated positions. In both cases, we propose the corresponding collision-resolving mechanisms by methodically recapturing the colliding time slot or acquiring the available free time slots after identifying the access vehicles and comparing the received signal strengths.

### 4.1. Cooperative Forwarding Mechanism

Each vehicle is considered to have a vehicle information table (VIT) about itself and its one-hop neighboring vehicles, which comprises information such as vehicle identification (ID), reserved time slot ID, vehicle position, vehicle speed, moving direction, and channel number. To join the channel, all vehicles need to sense for one frame period to identify which time slots have not been occupied. Additionally, then, they randomly select one of the time slots and send an updated info (UI) packet through the acquired time slot. By sensing the UI packet, every vehicle maintains the VIT of the one-hop set (1HS), two-hop set (2HS), and three-hop set (3HS) with neighboring vehicles, as illustrated in Table 2. It shows the structure of the vehicle information of the 1HS, 2HS, and 3HS with adjacent vehicles.

At first, any vehicle which belongs to the transmission range of *D*_SD_ is primarily chosen as a potential cooperative vehicle V_C_ at any moment the source vehicle V_S_ intends to transmit data packets. When the data packets with the cooperation header are received by the cooperative vehicles V_C_, they keep track of the channel and monitor the number of packet transmissions. Next, cooperative transmissions will initiate if V_C_ can confirm that its received SINR from the destination vehicle V_D_ is greater than the preset threshold *β*. Afterward, V_C_ determines the successful transmission rate between *V*_D_ and itself by means of observing the previously received data packets. Meanwhile, V_S_ adds cooperation header information to the packet including the number of allocated time slots, available free time slots, and the IDs of potential cooperative vehicles. V_S_ also determines the minimum number of cooperative vehicles to be deployed for the whole transmission range in order to minimize transmission overhead.

#### 4.1.1. Optimum Cooperative Vehicle Determination

If there is more than one V_C_ within the transmission range of *D*_SD_, it is necessary to select the most efficient cooperative vehicle to reduce unnecessary transmission times and the concerned link interference [51,52]. The precise procedure for choosing the optimum cooperative vehicle is covered in this part. When the vehicles are randomly dispersed along a multilane roadway, we first consider the traffic scenarios of cooperative vehicles V_C_ before and after a particular period *t*_β_.

As shown in Figure 4a, when the vehicles are moving in the right direction of the roadway before a *t*_β_, the communication radius *R*_c_ of the source vehicle V_S_ can cover vehicles V_1_, V_2_, V_3_, and V_4_; hence, the multi-paths based on these four vehicles are considered to forward the data packets to the destination vehicle V_D_ and they are generally considered as potential cooperative forwarding vehicles in this network. In Figure 4a, two vehicles, V_1_ and V_3_, are considered not effective for forwarding the data packets in this network scenario and they are defined as only suitable as relay vehicles V_R_. The reason is that even though they are located in the *R*_c_ of V_S_, but not in the direct communication range of V_D_, they also need to communicate with vehicles V_2_ and V_4_ to relay the data packets to V_D_.

On the other hand, vehicles V_2_ and V_4_ are in the intersection area *R*_S-D_ (i.e., the area between the direct communication range of V_S_ and V_D_). Therefore, V_2_ and V_4_ are successfully selected as V_C_ and added to the list of the VIT by defining a set of V_C_ in order to avoid excessive cooperative transmissions. Eventually, the vehicle which has the closest position with *D*_SD_ is selected as the optimum cooperative vehicle V_OC_. The verification process of a new vehicle V_N_ transforming into V_OC_ is depicted in Figure 5.

It is obvious that the successful forwarding rate reaches the maximum transmission rate when the optimum cooperative vehicle is selected accurately. As a first step, the distance of the potential cooperative vehicle from V_S_ is defined as *D*_C_ and it can be calculated as:(3)$$D_{C}=\sqrt{{\left(x_{c}-\frac{x_{s}+x_{d}}{2}\right)}^{2}+{\left(y_{c}-\frac{y_{s}+y_{d}}{2}\right)}^{2}},$$
where $\left(x_{c},y_{c}\right)$,$\left(x_{s},y_{s}\right)$, and $\left(x_{d},y_{d}\right)$ are the corresponding coordinates of the potential cooperative vehicle V_C_, the source vehicle V_S_, and the destination vehicle V_D_, respectively. If $x_{s}\leq D_{c}\leq x_{d}$, the received signal strength *RSS* at V*_D_* can be defined as:(4)$$RSS_{V_{D}}=10n_{1}\log_{10}(\frac{D_{C}}{x_{c}})+P_{t}+\ell_{p}+X_{\sigma },$$
where $X_{\sigma }$ is a Gaussian-distributed random variable with zero mean and standard deviation σ. At this point, the value of the cumulative distributive function of $X_{\sigma }$ at the origin can be considered as having a change of 0.5.

In the next step, to select the most optimum vehicle from a number of cooperative vehicles, the vehicles within the lower values of DC are listed first in the VIT, and the vehicle with the closest position to the midpoint between V_S_ and V_D_ is successfully selected as the V_OC_. Then, in priority order, the other vehicles are put into the substitution list of the VIT in case urgent replacement of V_OC_ is required upon a rapid topology change.

When the mobility of the vehicles is changed to a new scenario after a particular period *t*_β_ as shown in Figure 3b, vehicle V_2_ leaves the *R*_S-D_, and vehicle V_5_ moves forward into the intersection area of *R*_S-D_, and it seems to perform as a V_C_ instead of the former vehicle V_2_. At the same time, from the opposite direction, vehicle V_11_ also moves into the *R*_S-D_, and another vehicle V_14_ enters into the *R*_c_ of V_S_. However, both V_11_ and V_14_ are excluded from the operation process of L-CCHI since they are assigned to the R-CCHI by taking advantage of the disjoint time slot assignment upon the opposite directional mobility of vehicle traffic.

#### 4.1.2. Packet Structure

Let us consider the case that if a new vehicle V_N_ needs to transmit, it senses the channel for a frame period and searches for a free time slot. When the new time slot is identified, it simply transmits a UI packet on the selected time slot (for example, time slot *τ*_j_). After successfully reserving, the vehicles that have already been allocated a time slot in that frame will observe it, and specify if they have “heard” vehicle V_N_ in time slot *τ*_j_. It is assumed that after a vehicle successfully reserves a time slot for data packet transmissions, the time slot will not be released until it departs the system or a collision occurs as a result of topology changes. Each time slot is divided into two sections: the control preamble and the data payload. As shown in Figure 6, the proposed packet structure is depicted as follows:Legacy short training field (L-STF): executed for packet sensing, time acquisition, automatic gain control, and coarse frequency correction.Legacy long training field (L-LTF): executed for pilot-based channel estimation, fine frequency correction, and fine symbol-timing offset correction.Legacy SIGnal (L-SIG): contains metadata from the received configuration packets, such as the modulation coding scheme (MCS), and the physical layer convergence protocol (PLCP) service data unit (PSDU) length.Service: to set up the data scrambler.PLCP service data unit (PSDU): includes actual user information.Tail: executed for terminating convolutional code.Padding: executed for ensuring an integer number of symbols.

For actual transmissions, orthogonal frequency division multiplexing (OFDM) is used with a total of 64 sub-carriers (SCs). Among these 64 SCs, 52 SCs are used for carrying data and pilot symbols, and the remaining 12 SCs are for null SCs that carry no data. Additionally, the null SCs occupy the central 11 SCs and the 0th SC, and the pilot symbols occupy 4 SCs (Nos. 7, 10, 44, and 58). The remaining 48 SCs are used for data and the actual length of the data depends on the choice of MCS with the supported schemes. The frame information (FI) in the PSDU is organized as follows:Source temporary identifier (STI)—8 bits: the label of the vehicle “heard” by vehicle *V*_N_ on time slot *τ*_j_.Priority status field (PSF)—2 bits: the field indicates the priority of data transmitted in the time slot.BUSY—1 bit: a flag indicating whether time slot *τ*_j_ is free (0) or busy (1).File transfer protocol (FTP)—1 bit: executed in point-to-point transmission.

For local topology discovery, each vehicle sends all the information in its UI packet, and all adjacent vehicles update their VIT based on their received UI packets. If a vehicle does not receive any UI packets from its one-hop neighboring vehicles during three consecutive UI periods, it is assumed that all its neighboring vehicles have left its communication range and it updates its own VIT to announce temporarily losing contact with them.

Since preambles are deployed to identify a message and reservation collisions, in the CFC-MAC protocol, four types of packets are defined based on the different preambles: (1) a UI packet for time slot reservation, channel access, and periodic updates; (2) a data dissemination (DD) packet for transferring data and detecting collisions; (3) a successfully reserved (SR) packet for resolving collisions and recapturing a new time slot; and (4) an individual relay (IR) packet for relay vehicles to announce their exclusion from the cooperation process. Originally, each vehicle periodically transmits a UI packet for short status messages to notify the neighboring vehicles of its presence. Since SMs have two types (periodic beaconing messages and emergency event messages [53,54]), the UI packet and DD packet are mostly employed to enable periodic updates and data packet transmissions, respectively.

### 4.2. Vectorized Trajectory Estimation Mechanism

In VANETs, vehicles may frequently converge and diverge due to their varying velocities and routes. Merging collisions occur when two vehicles in different hop sets (HSs) accessing the same time slot become members of the same HS due to rapid changes in their positions after a particular period. As a result of merging collisions, there are subsequent access collisions between the vehicles that have abandoned their time slots [31]. In this subsection, we first examine the merging collision recognition procedure, and then, propose an SINR-based estimation mechanism to completely eliminate them.

Generally, merging collisions are likely to occur in the following two cases: (1) vehicles moving in the same direction with different speeds, and (2) vehicles moving in opposite directions on a bi-directional road [29]. Since our proposed system model only utilizes the disjoint time slot assignment by dividing L-CCHI and R-CCHI for the left and right directions, it is not necessary to consider merging collisions that occur on a bi-directional road. Therefore, we specifically emphasize the merging collisions of vehicles moving in the same direction in this paper.

As shown in Figure 7, two vehicles, V_S1_ and V_D1_, in a 2HS move at a higher speed than three vehicles, V_S2_, V_OC_ and V_D2_, in their three-hop sets (3HSs). Concurrently, V_D1_ and V_D2_ reserve the same time slot when they drive into direct communication range of each other after a period of *∆t*; this conflict of using the same time slot leads to a merging collision.

The residence time of a vehicle in a service area can be calculated by using mobility metrics, such as position, direction, and speed, to synthesize the transmission probability of each individual vehicle with respect to its residence time [55]. In this paper, we propose a VTE mechanism, in which the relative velocity of vehicles with different speeds is mainly considered to reduce merging collisions. Let us consider that each vehicle can be aware of its own information regarding velocity $v_{t}$, acceleration $a_{t}$, and current position $\phi_{t}$ at time *t*, which are vectorized in *x* and *y* coordinates. For a moving vehicle, after a given period *ξ*, the current moving position $\phi_{t}$ can be calculated as
(5)$${ф}_{t}=\frac{a_{t}\xi^{2}}{2}+v_{t},$$
and let us assume that $\phi_{t-\xi }$ is the previous moving position at time t−ξ. Next, the current moving direction $\beta_{\xi }$ is given by
(6)$$\beta_{\xi }=cos^{-1}\frac{{ф}_{t-\xi }{ф}_{t}}{\left|{ф}_{t-\xi }\right|\left|{ф}_{t}\right|}.$$

If a merging collision occurs in an HS, firstly, V_S_ needs to compare the relative velocities of the collided vehicles before the time interval *∆t* by identifying the previous position $\phi_{t-\xi }$ in this HS. If no vehicles have a higher velocity, the time interval is expanded into 2*∆t*, 3*∆t*, 4*∆t*, etc., and the previous velocities are compared. If it is discovered that one of the collided vehicles has a higher velocity at the previous positions, that vehicle is put into a blacklist for a random time interval since it is considered to enter into this HS from other HSs. This random time interval is based on the priority order of the arbitrary interframe space (AIFS). A message has a better chance of being transmitted with minimal latency when its AIFS period is shorter, which is extremely important for delay-critical applications. After waiting for its allocated time period, in the next frame, the collided vehicle with higher velocity can randomly pick up an available time slot from the remaining free time slots, and it is required to send its SR packet to notify other vehicles that the merge collision is fully resolved. Meanwhile, the other collided vehicle with lower velocity is required to recapture the colliding time slot again, and also needs to send its SR packet. Therefore, merging collisions are completely eliminated.

### 4.3. Heterogenous Collision Resolution

The proposed CFC-MAC protocol utilizes multiple cooperative vehicles to forward data packets, and then, intentionally selects the optimum cooperative vehicle so as to improve the probability of successful packet transmissions. Based on the road connectivity probability under high network load [47,56], the existing TDMA-based MAC protocols do not show compatibility with the occurrence of multiple cumulative collisions in stochastic traffic conditions. Therefore, unified models for defining heterogeneous collisions become captious for traffic characterization and modeling. In this subsection, we develop sustainable collision models by focusing on the context of typical and aggregated heterogeneous collisions under congested or heavy traffic flow.

If a vehicle has been discovered in the occurrence of any type of collision, it is unable to recapture the same time slot in the current frame and needs to acquire a new time slot in the next frame. In this case, it is extremely important that the new time slot must be completely free and not occupied by any other vehicles in the upcoming frames to eliminate the possibility of further subsequent collisions.

#### 4.3.1. Typical Heterogenous Collision Resolution

The time slot allocation procedures of typical heterogenous collisions in our proposed CFC-MAC protocol are shown in Figure 8. First, we consider that all vehicles are moving in the right direction, as illustrated in scenario A. It is clear that a new vehicle V_N_ is not in the *R*_S-D_; therefore, only one time slot for V_S_ is available in frame 0. Considering scenario B, the network topology is changed when V_N_ enters the *R*_S-D_. In the case of V_N_ attempting to join the network, it first senses the channel for one frame (beaconing time) to figure out which time slots have not been occupied in this frame. Next, V_N_ randomly chooses one of the available time slots to reserve the channel and becomes a V_OC_ after being fully selected by following the procedures shown in Figure 5. In scenario B, another new vehicle also goes into the *R*_c_ range of V_S_, but it can only be defined as a V_R_ because it cannot directly communicate with V_D_. As illustrated in the frame (*n-b*), V_OC_ and V_R_ can send their UI packets to their successfully reserved time slots, respectively.

Scenario C comes into existence when the network topology is changed to be broader than the 2HS of the V_OC1_ and V_OC2_, and they are successfully involved in the whole process of data transmission from V_S_ to V_D_. Meanwhile, vehicle V_R_ is also entering into the transmission range between V_OC2_ and V_S_, under the same condition as V_OC1_. In this case, cooperation collision occurs when vehicle V_R_ is acting as a cooperating vehicle for relaying the lost messages and trying to reserve the allocated time slot, whereas the vehicles V_OC1_ and V_R_ are contending to reserve the same time slot (the seventh time slot) as depicted in the frame (*n*
**−***)*. As a result, these two vehicles cannot affiliate with the current frame.

To handle the issue, both collided vehicles have to wait for the next frame (*n*
**−**
*d + 1*) and sense the channel again. In the (*n*
**−**
*d + 1*)th frame, V_S_ needs to send a DD packet to its own time slot with the preamble part containing the information about V_R_, stating that it must put the colliding seventh time slot into the blacklist for a random period based on AIFS. After waiting for this period, V_OC1_ can recapture the seventh time slot again and it needs to send the SR packet to notify other vehicles that the collision is fully resolved. On the other hand, another collided vehicle, V_R_, can randomly pick up an available time slot from the remaining free time slots in the frame, and it is also required to send the SR packet. Consequently, the cooperation collisions have been resolved, and all collided vehicles can pick up the available time slots and continue the cooperating process.

Scenario D is incurred when two V_OC_ vehicles within the direct communication range of the 2HS are contending to reserve the same time slot, which directly causes access collisions. To resolve this, the received SINR values of these two collided vehicles are compared first, and the vehicle with a greater received SINR value is selected as the front vehicle, here V_OC1_ is selected in this demonstration since the relative distance is directly proportional to the received SINR. Then, V_S_ also needs to send a DD packet to its own time slot, whereas its preamble part contains instructions stating that the preceding vehicle *V*_OC1_ can recapture the colliding time slot (the second time slot) and the following vehicle V_OC2_ is required to wait for a random period depending on AIFS. Subsequently, V_OC2_ can acquire a new available free time slot and sends the SR packet, acting as the same foregoing procedure of resolving cooperating collisions.

To examine the packet transmission collisions of the relay vehicles, the IR packet is sent out by a relay vehicle when it has successfully reserved the time slot to inform other types of vehicles that it is not involved in the cooperation process. Upon receiving the time slots with the IR packet, a relay collision between the relay vehicles can also appear, as depicted in Scenario E. To address this collision, a DD packet is employed by V_S_ to place these collided vehicles in the exclusion list for a random amount of time in the next frame (*n*
**−** *h* + 1). After waiting for this random amount of time, both collided relay vehicles can join the channel again by sensing and acquiring the new time slots based on the prior order in comparing the intensity of their received SINR, and both of them also need to send SR packets, respectively. In this way, relay collisions are successfully mitigated.

Without the occurrence of access collisions, a collision can nevertheless occur, rarely, when two vehicles unexpectedly reserve the same time slot, and this kind of collision is called a reservation collision. To tackle this issue, both collided vehicles are put into the exclusion list at first; then, the received SINRs are compared, and V_S_ will send its DD packet including the instruction that the vehicle with the greater SINR has a higher opportunity to recapture the time slot in the next frame, and the other collided one can attempt to join the channel after waiting for a random period. Both of them also need to send their SR packets after reserving the available time slots successfully to announce that the access collisions have been resolved.

Furthermore, failure in packet reception may happen due to a transmission error or interference when different packets are assigned to the same time slot. In this case, vehicle V_S_ will detect the colliding time slot with the packet failure and put it into the exclusion list for a random period. Collision information in the DD packet is released in the next frame to reassign a new free time slot again, and the collisions are resolved. Through this approach, subsequent collisions can be discovered and efficiently mitigated. However, there is still the possibility of encountering aggregate heterogeneous collisions, which happen when numerous collisions take place at once because of heavy or highly congested traffic load.

#### 4.3.2. Aggregated Heterogenous Collision Resolutions

Very little research has been focused on modeling in the context of aggregated heterogeneous collisions, particularly in situations under highly congested or heavy traffic flow. Therefore, we develop sustainable collision models when making decisions on discovering efficient solutions to resolve these types of collisions. In this subsection, we first discuss the case of aggregated heterogenous collisions of hidden terminals (HTs) and exposed terminals (ETs) at dissociated positions, occurring simultaneously but quite independently. Additionally, then, the case of their aggregated heterogenous collision at an associated position is presented. In both cases, we provide the corresponding resolving methods depending on their received SINRs, vehicle mobility, and relevant collisions.

1.Aggregated Heterogenous Collisions of Hidden and Exposed Terminals at Dissociated Positions

When two vehicles are not within direct communication range and contend to reserve the same time slot, it leads to a hidden-terminal problem. As seen on the left of Figure 9, vehicles V_S1_, V_D1_, and V_OC1_ can successfully reserve time slots 2, 6, and 8, respectively. However, vehicle V_OC1_ is entering the intersection area between V_S1_ and V_D1_; as a consequence, V_S1_ and V_D1_ cannot sense each other, while V_OC1_ can sense both of them. In this instance, vehicles V_S1_ and V_D1_ unexpectedly reserve the same time slot (the third time slot) and an HT occurs.

In the meantime, as seen on the right in Figure 9, when two sender vehicles (V_S2_ and V_S3_) are within communication range of each other, and their expected destination vehicles (V_D2_ and V_D3_) are far away from each other, respectively, V_S2_ and V_S3_ can concurrently transmit the message to V_D2_ and V_D3_, respectively, using the same time slot without any conflict. However, V_S2_ and V_S3_ are not allowed to reserve the same time slot (the seventh time slot in Figure 9), and the ET problem occurs. This kind of scenario is termed aggregated heterogenous collisions at the dissociated position. From the above phenomenon, we have discovered that both HT and ET problems can lead to inefficient reusability and directly cause time slot collisions.

In order to resolve an HT problem, at first, the vehicle between the two collided vehicles is considered to be an access point vehicle V_ap_ since it is the only vehicle that can discover the current collision. Alternatively, to resolve an ET problem, the two collided vehicles (V_S2_ and V_S3_) can be assumed to be V_ap1_ and V_ap2_, respectively, since both of them are within each others’ communication radii.

The procedures of the slot reallocation mechanism for aggregated heterogenous collisions of HTs and ETs at dissociated positions (SRM-DP) are shown in Figure 10 for resolving any HT or ET problems. First, it is necessary to confirm that the number of collided vehicles *N*_cv_ is less than the number of currently available free time slots *N*_fs_. If not, the vehicle is required to wait for the next frame. If yes, the V_ap_ compares the received SINRs of the collided vehicles through VIT. To resolve the HT problem, the vehicle with the higher received SINR is given priority to recapture the colliding time slot since it is assumed to get closer to V_ap_ than the others.

To resolve the ET problem, both V_ap1_ and V_ap2_ need to compare the received SINRs of their adjacent vehicles (V_D2_ and V_D3_), respectively, and the V_ap_ with the higher received SINR with its adjacent vehicle is selected to recapture the colliding time slot. Then, the other residual vehicle needs to acquire any of the other free available time slots. After both collided vehicles have successfully reserved the available time slots, respectively, they need to broadcast SR packets to notify other vehicles. By following this process, the collisions due to HT and ET problems are successfully eliminated.

2.Aggregated Heterogenous Collisions of Hidden and Exposed Terminals at Associated Positions

If multiple aggregated collisions of HTs and ETs have occurred jointly and simultaneously, it is necessary for all collided vehicles to acquire a new time slot in the next frame, whereas the collided vehicles with higher received SINRs can recapture the colliding time slots. As shown in Figure 11, we consider that three types of collisions occur concurrently, with one collision occurring due to ETs, and the other two collisions occurring due to HTs; therefore, a total number of four HTs and three ETs occur in this network scenario.

The procedures of the slot reallocation mechanism for aggregated collisions of HTs and ETs at associated positions (SRM-AP) are discussed in Figure 12. First, it is necessary to confirm that *N*_cv_ < *N*_fs_. If not, it is necessary to wait for the next frame. If yes, we compare the number of V_ap_ vehicles within the collisions. If there is only one V_ap_, we can follow the SRM-DP procedure. If there are more than two V_ap_ vehicles, the received SINRs of both are compared, and the V_ap_ with the highest SINR value related to the HTs is given higher priority.

On the basis of Figure 11, three vehicles (V_S1_, V_S2_, and V_S3_) are within the communication range of HT-1 and HT-2. Therefore, the received SINRs from these two HTs are compared first, and then, the V_ap_ with the higher SINR is selected to resolve the collisions. Then, the SINRs of the collided vehicles are compared. Moreover, the collided vehicle with a higher SINR is selected to recapture the colliding time slot since it is anticipated to approach the targeted V_ap_. Next, the rest of the vehicles can acquire any of the remaining available free time slots. After successfully reserving their respective time slots, the two vehicles also need to transmit an SR packet to inform the others, respectively.

To resolve the aggregated collisions of ETs, as shown in Figure 12, all V_ap1_, V_ap2_, and V_ap3_ vehicles need to compare the SINRs of their adjacent vehicles, including V_D1_, V_D2_, V_D3_, and *V*_D4_, respectively, and the V_ap_ with the higher received SINR relative to its adjacent vehicles is selected to recapture the colliding time slot. Then, the same steps with the aforementioned procedures are followed. By following this SRM-AP mechanism, the cumulative collisions due to the aggregated heterogenous collisions of HT and ET problems are successfully eliminated.

## 5. Performance Analysis

In this section, we first investigate the performance analysis of collision probability and packet reception probability to evaluate the protocol performance and optimize network parameters. Then, we analyze access delay to validate the protocol’s fairness and efficiency.

### 5.1. Collision Probability

In this subsection, we investigate the average number of all types of collisions experienced by the CFC-MAC and derive the probability mass function (PMF) to describe the discrete probability distribution of these collisions. Let us consider that the vehicles are distributed randomly on a multi-lane roadway and follow the Poisson distribution [48] for a limited transmission range. Traffic density $\lambda_{T}$ can be acquired through fixed installed detectors, such as cameras, inductive loops, microphones, and Electronic Toll Collection (ETC) readers [46,57]. The PMF of *n* vehicles, running on a road segment of length $L_{r}$, can be expressed as follows:(7)$$P(n)=\frac{{\left(\lambda_{T}L_{r}\right)}^{n}e^{-\lambda_{T}L_{r}}}{n!}.$$

When at least two vehicles access the same time slot, there is a probability of a collision. Different kinds of collisions may occur within a given frame when two or more vehicles are contending to reserve specific time slots. Therefore, the probability that a collision happens at a specific time slot is 1−P(n). Let j be the number of time slots among *i* encountering collisions whereby j≤n/2; the probability P(j) can be defined as
(8)$$P(j)=C_{i}^{j}{\left[1-P(n)\right]}^{j}P{\left(n\right)}^{i-j},$$
and the average number of collisions $N_{c}$ can be calculated as
(9)$$N_{c}=\sum_{j=1}^{n/2}jP(j).$$

Let us assume that a frame has $n_{\tau }$ number of time slots for each direction and each vehicle needs to assign a certain time slot where $n_{\tau }\geq n$. In general, there are at least two vehicles for one colliding time slot, and collided vehicles are also able to encounter a new collision again when they try to acquire a new time slot. In this case, the same process of achieving achieve $N_{c}$ can be iterated until no collisions are left. Then, the probability of *x* number of collisions for $n_{\tau }$ number of time slots can be defined as
(10)$$P(x)=\frac{{\left(N_{c}n_{\tau }\right)}^{x}e^{-2N_{c}N_{\tau }}}{x!}.$$

### 5.2. Packet Reception Probability

To validate our design process, an analytical expression of packet reception probability is necessary. Considering that a packet’s arrival at a vehicle follows the Poisson point process (PPP), in each frame, the probability that a specific vehicle generates a packet is $p=1-P_{r}(0)$, where $P_{r}(0)$ represents the probability of generating a zero packet. When there are *n* vehicles in the system, the probability that *x* cooperative vehicles will generate a packets is
(11)$$P(x)=C_{n}^{x}p^{x}{\left(1-p\right)}^{n-x},$$
where $C_{n}^{x}=n!/(x!(n-x)!)$. Let us assume that there is $N_{oc}$ number of optimum cooperative vehicles; the probability that the cooperative vehicle receives *k* packets is
(12)$$P\left\{C=k\right\}=\sum_{x=k}^{n}P(x)C_{x}^{k}{\left[{\left(1-P_{re}\right)}^{N_{oc}}\right]}^{k}{\left[1-{\left(1-P_{re}\right)}^{N_{oc}}\right]}^{x-k},$$
where $P_{re}$ is the probability of the relay vehicle receiving the packets and $C_{x}^{k}=x!/(k!(x-k)!)$.

### 5.3. Access Delay

In the VANET environment, the access delay is defined as the time duration needed by a new vehicle to access a time slot successfully for data packet transmissions, and it can be disastrous in time-sensitive deployments where immediate detection and actions impact security, safety, and link failures. When a vehicle attempts to access a time slot, it will receive a DD packet with access denial or access failure from its neighboring vehicles if another vehicle is exploiting it. In our CFC-MAC, we consider the access failure for a transmission range based on the received SINR threshold.

The maximum density achieved under congestion, called the jam density $K_{c}$, within a length of a roadway $L_{r}$ at a given time is equal to the inverse of the average spacing of the *n* vehicles, i.e.,$K_{c}=S_{n}/L_{r}$. For the transmission range *D*_SD_, the average number of vehicles *M*_v_ for *l*_s_ lanes can be stated as follows:(13)$$M_{v}=l_{s}K_{c}D_{SD}.$$

For a time duration $t_{d}$, the average velocity of a vehicle $v=2R_{c}/t_{d}$. When $N_{\tau }$ number of total time slots have j number of colliding time slots, the successful channel access of 1HS, 2HS, and 3HS is $n_{\tau }-j/n_{\tau }$, $n_{\tau }-2R_{c}M_{v}/n_{\tau }$, and $n_{\tau }-3R_{c}M_{v}/n_{\tau }$, respectively. The access failure probability of the 3HS range can be defined as
(14)$$P_{failure}=\sum_{j=0}^{n/2}\left[1-\left(\frac{n_{\tau }-j}{n_{\tau }}\frac{n_{\tau }-2R_{c}M_{v}}{n_{\tau }}\frac{n_{\tau }-3R_{c}M_{v}}{n_{\tau }}\right)\right]P(x).$$

By following the geometric distribution theory, if the probability of successful transmission in each trial is $P_{failure}$, the probability of possessing successful channel access after *X* failed for the *k*^th^ trial (out of finite trials) is
(15)$$P\{X=k\}=(1-P_{failure})P_{failure}^{k-1},$$
where *k* = 1, 2, 3, 4, … and the expected value E(*X*) is $1/P_{failure}$. When a vehicle wants to utilize a time slot, it initially looks for free time slots for one frame period. After reserving a time slot, the UI packet is transmitted, and the vehicle will be informed as to whether it may utilize the reserved time slot after a certain amount of time $t_{\omega }$. If not, it will attempt to acquire another time slot. Let us assume that $T_{co}$ is the average time between a cooperative vehicle receiving an SM from the previous source and rebroadcasting it, $T_{ps}$ is the average time taken for a collided vehicle to recapture the colliding time slot again, and $T_{ns}$ is the average time taken for a collided vehicle to reserve a new time slot. Therefore, the total average time taken for the whole process $t_{\omega }$ will be
(16)$$t_{\omega }=\sum_{j=0}^{n/2}T_{co}+T_{ps}+T_{ns},$$
where $T_{ps}=\tau_{\min}N_{\tau }$, and $\tau_{\min}$ is the minimum time interval of a time slot based on the frame size of a packet. The time interval between receiving the notice and moving on to the next attempt (including the initial time slot decision taken after scanning the available time slots) is a random variable *z* following a Uniform Cumulative Distribution Function (UCDF),$z\sim U(0,t_{\omega })$, where the mean (first moment) of the distribution is $\mu =t_{\omega }/2$. Based on this approach, the access delay $D_{A}$ can be calculated as
(17)$$D_{A}=\frac{1}{1-P_{failure}}\left(t_{\omega }+\mu \right)+t_{\omega }.$$

It is worth mentioning that a more precise access delay can be achieved depending on the access failure probability within the transmission range.

## 6. Performance Evaluation

In this section, we first elaborate on the simulation parameters and performance metrics; then, we evaluate the performance of our proposed CFC-MAC in terms of five key measures: access delay, packet loss ratio (PLR), throughput, end-to-end (E2E) delay, and collision rate. Using consistent simulation settings for the highway traffic scenario, the performance of the proposed CFC-MAC protocol is evaluated in comparison with the EVC-TDMA [28], ECTOB-MAC [43], and PeerProbe-MAC [46] protocols.

### 6.1. Simulation Parameters

We performed extensive simulations in OMNET++5.6.2 with the INET 4.2.8 framework for the event-based network simulation, which combines open-source vehicular network simulator VEINS 5.2, and Simulation of Urban MObility (SUMO) 1.11.0 msi (eclipse) for the generation of real-world mobility models, which include the deployment of the Intelligent Driver Model (IDM) and the Minimizing Overall Braking Induced by Lane Changes (MOBIL) model. We obtained the randomly partitioned map from OpenStreetMap (OSM) and turned it into a SUMO simulation using INET and VEINs together to simulate a V2V network. In the simulation, the radio channel models, such as shadowing and fading effects [47], were not taken into account.

A 5 km bidirectional road is set with four lanes in each direction and varying speed limits of 60, 80, 100, and 120 km/h, and the total network area is 6 × 4 km^2^. The executed vehicles in the traffic scenarios have distinct performance characteristics, such as length, acceleration, deceleration, and maximum speed. The attributes of vehicles utilized in the SUMO are summarized in Table 3. The vehicle performance parameters and destination are chosen randomly when they enter the simulation system, and they leave the system when they reach the chosen destination.

The simulation of CFC-MAC is implemented as follows. Each vehicle maintains a VIT status for all time slots, and its operation is dependent on the VIT information. The deployment of multi-channel devices allows the vehicles to listen to all channels and broadcast SM at the selected time slot. The PHY layer of CFC-MAC is modeled in the same way as IEEE 802.11p, in which the protocol packet data unit (PPDU) [58] is composed of a preamble, a signal field, and a payload component containing the useful data. A total of 52 subcarriers (data + pilot) are assigned numbers from −26 to 26, and the symbol interval is 8 μs, respectively. On the other hand, the EVC-TDMA, ECTOB-MAC, and PeerProbe-MAC protocols are implemented based on the existing INET framework with a data rate of 6 Mbps and a communication radius of 250 m.

The packets are successfully received by a vehicle if the received SINR is more than the preset threshold *β*, and a collision occurs when there is more than one transmitter using the same time slot within the 2HS range of the receiver. By considering the antenna gains and feeder losses, it is possible to determine the transmission power for various vehicle densities, as shown in Table 4—with the results for the number of optimum cooperative vehicles $N_{oc}=10$—by using (2) based on the desired SINRs of 5, 20, 40, and 60; these correspond to time slot duration of 4 ms, 2 ms, 1 ms, and 0.5 ms, respectively. Since a signal with an SINR value of 20 dBm or more is recommended and the duration of a time slot should not be longer than 2 ms, it can be seen that λ < 0.059 and the transmitted power is less than 25 dBm. Furthermore, the network simulation parameters are specified in Table 5.

### 6.2. Performance Evaluation

In the implementation of all the compared MAC protocols, a cooperative vehicle attempts to forward the data packet a specified number of times, regardless of receiving a packet from a farther vehicle or source vehicle. Therefore, the packet reception probability is effectively improved by increasing the maximum number of transmission counts. On the contrary, the relay vehicle aborts its forwarding process upon receiving the packet, which can definitely reduce the number of retransmission attempts. To evaluate the performance of our proposed protocol under varied traffic densities, the following is a list of definitions for the evaluated performance metrics:Access delay: The time duration required by a new vehicle to access a time slot successfully for data packet transmission. It depends on the number of total vehicles when a vehicle has acquired a time slot and utilizes it in all continuous frames without collision.Packet loss ratio: A packet loss denotes that a packet is not received because of a collision or channel fading. The packet loss ratio (PLR) can be defined as
(18)$$PLR=1-\frac{\#\mathrm{of}\;\mathrm{received}\;\mathrm{packets}}{\#\mathrm{of}\;\mathrm{expected}\;\mathrm{packets}\;\mathrm{to}\;\mathrm{be}\;\mathrm{received}}=1-\eta .$$

Throughput: The average number of successful data transmissions per unit of time as a fraction of the channel capacity. It is controlled by bandwidth, the received SINR, and hardware limitations.End-to-end (E2E) delay: The time taken for a packet to route through the network from a source vehicle to its destination vehicle.Collision rate: The average number of collisions per frame. The collision rate significantly depends on traffic density and the total number of available time slots.

For various vehicle densities, Figure 13 contrasts the simulation results of the access delay with the analytical results, whereas the power control mechanism is not deployed to provide a fair assessment. The simulations were performed ten times, and the standard deviations and average values were plotted on a graph. The derived access collision probability and access delay are estimations. Therefore, the analytical results provide an upper bound for the access delay. In the simulation, there is only a minor difference between the bound and the simulated access delay, and this difference tends to increase when the density is very high and the frequency of cumulative heterogenous collisions is significant. As a result, the proposed CFC-MAC is adaptable and capable of ensuring an effectively consistent access delay up to a specific number of vehicles.

Figure 14 presents the analytical results for access delay related to different numbers of optimum cooperative vehicles V*_OC_*. With the increase in vehicle densities, a growing number of optimum cooperative vehicles participate in the communication process. A total of 184 available time slots is set, and each time slot has a duration of 0.5 ms. The access delay slightly increases and is on the verge of 0.5 s, even when the vehicle density goes up to 300. Moreover, it can be seen that there is a significant difference at the beginning of the access delay with the enlarging of vehicle densities. However, higher vehicle densities do not have a substantial impact on the access delay, showing that it is almost stable and deterministic. The reason is that CFM can reduce excessively long retransmission delays and endure the occurrence of heterogenous collisions in highly congested traffic scenarios.

In Figure 15, we compare the packet loss ratio (PLR) between our CFC-MAC and the aforementioned three protocols at 0.4 and 0.7 percent of *P*_k_, which is the probability of the cooperative vehicle receiving *k* packets, and overall, the results for 0.7 are better than those for 0.4. We can observe that all the MAC protocols provide a low PLR under different traffic densities. When the background traffic level increases, the PLR also becomes steadily higher. The proposed CFC-MAC yields the lowest PLR among the four different MAC protocols because cooperative forwarding with a selected number of V*_OC_* vehicles can reduce the number of unnecessary retransmissions of relay vehicles. It should be noted that the proposed CFC-MAC slightly underestimates the figure at *P*_k_ = 0.7, resulting in a smaller-scale increase in PLR than at *P*_k_ = 0.4 and a relatively lower result than the other protocols. This is because due to the probability simplification in the derivation through rapid identification of the next V*_OC_* after the previous forwarder, PLR is effectively minimized by manipulating the maximum number of transmissions.

Figure 16 depicts the normalized throughput for different protocols. From the figure, it can be seen that more packets result in larger throughput when there is little congestion. We implement the channel busy ratio of the ECTOB-MAC into the throughput calculation since it is the average value for the ratio of the channel busy time of each vehicle over the broadcast period of an emergency warning message (EWM). We can see that the proposed CFC-MAC protocol outperforms other competing protocols under diversified traffic densities. The prominent reason is that the CFC-MAC is capable of adapting to unbalanced traffic conditions and can reduce the various types of heterogenous collision, especially aggregated heterogenous collisions and same-direction merging collisions.

Figure 17 demonstrates how successfully CFC-MAC minimized the E2E delay in contrast to the other protocols. It can be clearly seen that it consistently achieves a comparable delay of no more than 8 ms at 350 vehicles/km, even at the value of *P*_k_ = 0.4. It is important to realize that EVC-TDMA has a substantial computation delay at a density of 200 vehicles/km, which is quite close to CFC-MAC’s E2E delay of around 5 ms; this is because a relay vehicle can straightforwardly find a sufficient number of cooperators with an EWM size of 500 bytes and significant background traffic levels. On the other hand, both ECTOB-MAC and PeerProbe-MAC have E2E delays above 10 ms and also reach steeper curves at higher densities. This is because the time slot partition method cannot provide enough time slots for vehicles on the dense side, which may result in various types of collisions.

In Figure 18, we compare the collision rated of four different protocols. Increasing the number of vehicles causes more contentious, noteworthy interference among vehicles and an excessive collision rate. The EVC-TDMA and PeerProbe-MAC have a steady collision rates with escalating density levels. Noticeably, the proposed CFC-MAC has the lowest collision rate and remains constant or slightly increasing, generally below 30% at both *P*_k_ = 0.4 and *P*_k_ = 0.7, whereas the number of collisions in a frame for ECTOB-MAC towers significantly at higher densities of between 140 and 160 at 350 vehicles/km. The main reason is that CFC-MAC’s VTE is more trustworthy in resolving same-direction merging collisions to a certain extent, and has a greater connection in cooperative forwarding schemes. Moreover, CFC-MAC can also extensively identify and reduce the occurrence of typical heterogenous collisions and aggregated heterogenous collisions, and gains more adaptability to real-world traffic scenarios.

## 7. Discussion

The existing TDMA-based distributive cooperative MAC protocols do not show compatibility with the occurrence of heterogenous collisions. Therefore, unified models of defining aggregated heterogeneous collisions become captious for traffic characterization and modeling in a congested state of traffic flow. The proposed CFC-MAC protocol utilizes optimum cooperative vehicles to forward data packets, so as to improve the probability of successful data packet transmissions and achieve deterministic access delay. Specifically, CFC-MAC can achieve 5% to 25% performance enhancement compared to other protocols.

## 8. Conclusions and Future Work

In this paper, we have presented a novel TDMA-based contention-free cooperative MAC protocol to develop a cooperative forwarding model by proficiently selecting the optimum cooperative vehicles. Based on the disjoint time slot partitioning method, same-direction merging collisions can be predicted and effectively eliminated using the SINR-based vectorized trajectory estimation method by identifying the upcoming pathways. Based on the proposed cooperative forwarding model, we scrutinize the diversified patterns of typical heterogenous collisions, including cooperation collisions, access collisions, relay collisions, and reservation collisions. Moreover, we investigate the aggregated heterogenous collisions of hidden terminals and exposed terminals at dissociated positions (which occur simultaneously but separately) and associated positions. By identifying the access point vehicles and comparing the received signal strengths, we provide pertinent resolving methods to reduce both types of heterogenous collision. Finally, we conduct extensive simulation experiments, and the results confirm the effectiveness of the proposed protocol. In the near future, we may study the effects of RSUs on cooperative forwarding mechanisms to select the optimum cooperative vehicle. Further, we intend to develop a machine-learning-based technique to resolve the various types of heterogenous collision for VANETs.

## Figures and Tables

**Figure 1 sensors-23-01033-f001:**
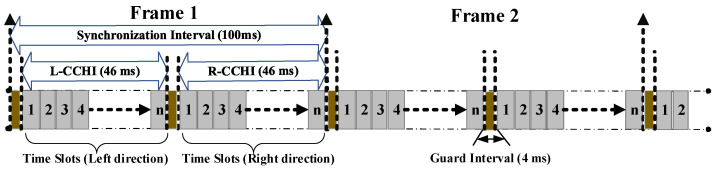
Each TDMA frame is divided into two sets of time slots: set L-CCHI for the left direction, and set R-CCHI for the right direction.

**Figure 2 sensors-23-01033-f002:**
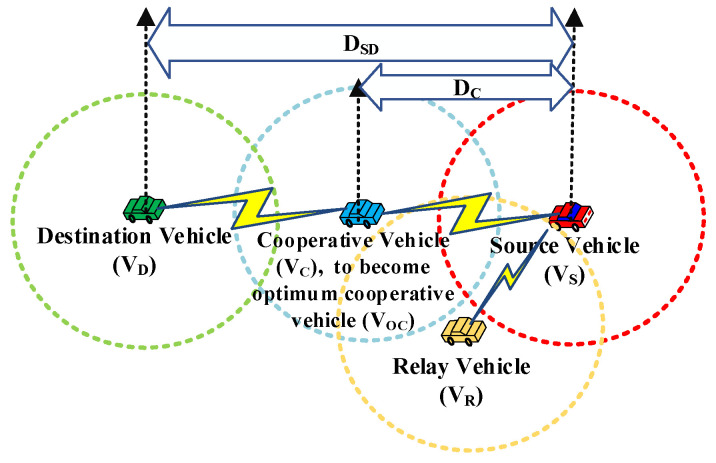
Cooperative relay between vehicles.

**Figure 3 sensors-23-01033-f003:**
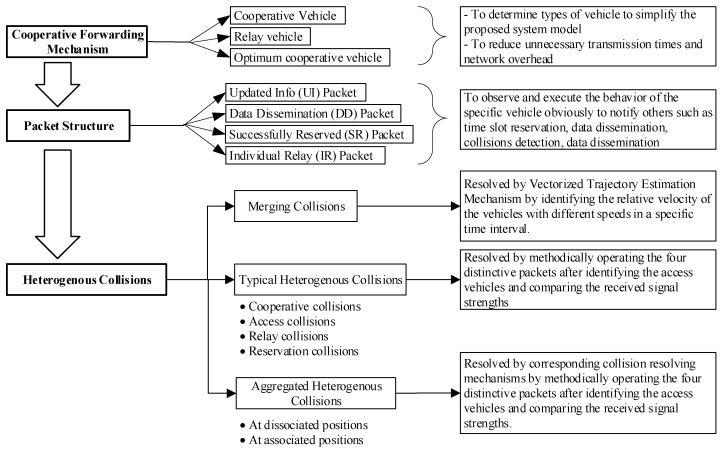
Structural flowchart of the proposed CFC-MAC protocol design.

**Figure 4 sensors-23-01033-f004:**
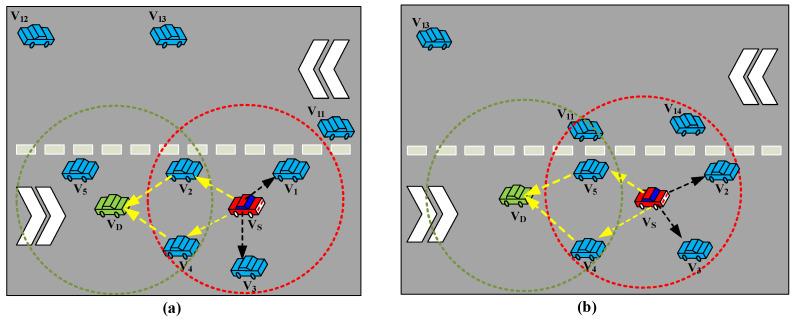
Cooperation models (**a**) before a particular period and (**b**) after a particular period.

**Figure 5 sensors-23-01033-f005:**
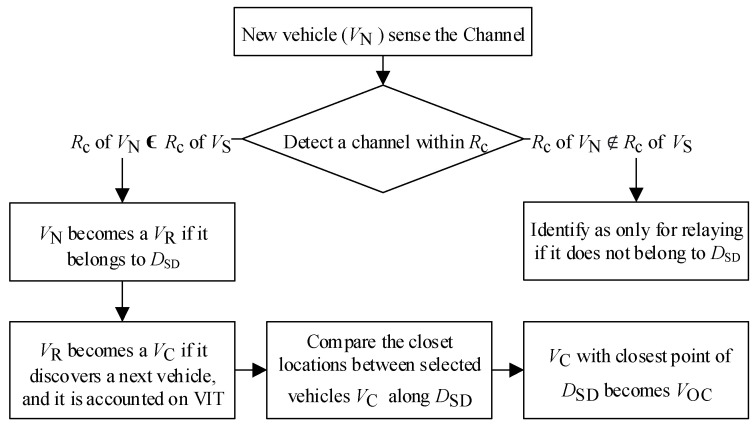
The verification process of optimum cooperative vehicle (V_OC_).

**Figure 6 sensors-23-01033-f006:**
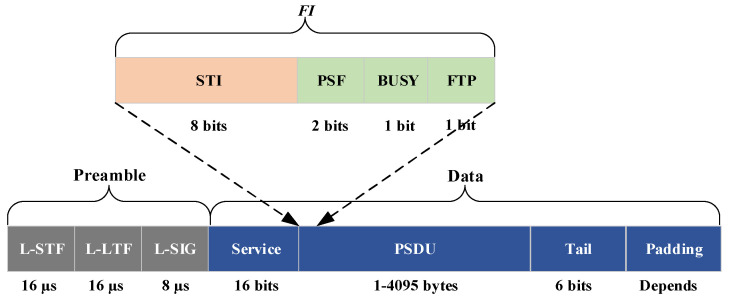
Packet structure.

**Figure 7 sensors-23-01033-f007:**
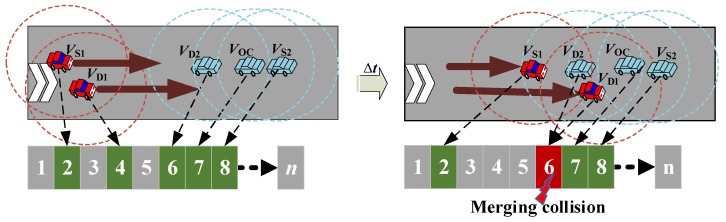
Merging collisions occur when vehicles move in the same direction at different speeds.

**Figure 8 sensors-23-01033-f008:**
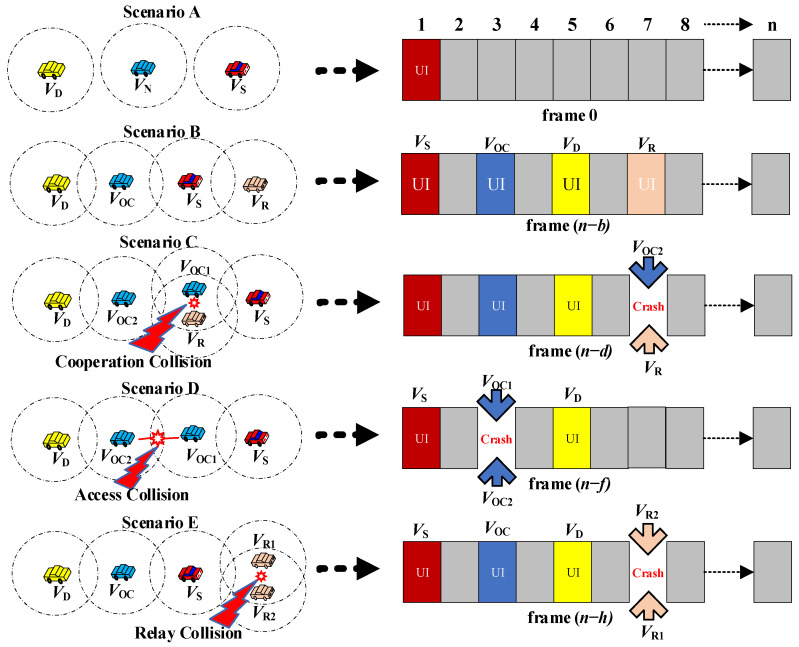
Time slot assignment procedures for resolving typical heterogenous collisions.

**Figure 9 sensors-23-01033-f009:**
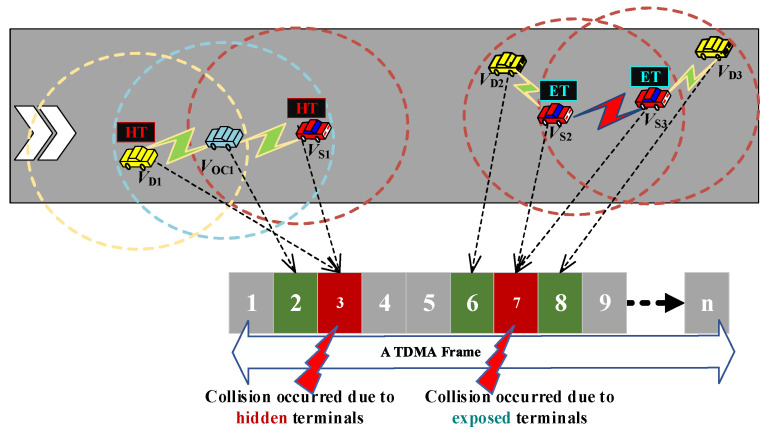
Aggregated collisions of HT and ET problems occurring at dissociated positions.

**Figure 10 sensors-23-01033-f010:**
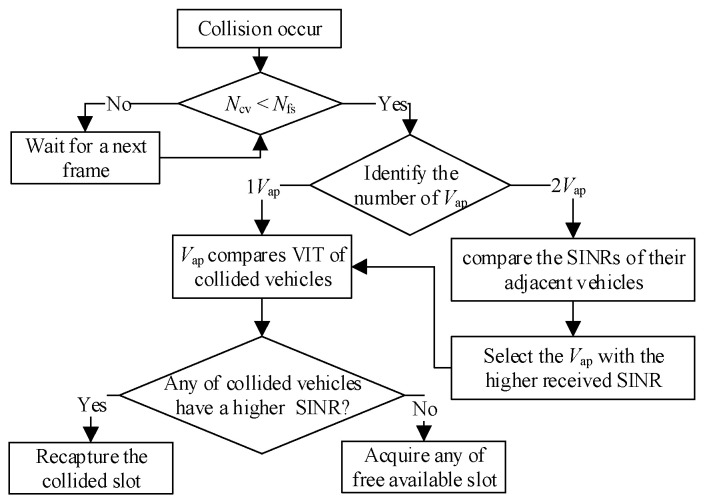
Slot reallocation mechanism for aggregated heterogenous collisions of HTs and ETs at dissociated positions.

**Figure 11 sensors-23-01033-f011:**
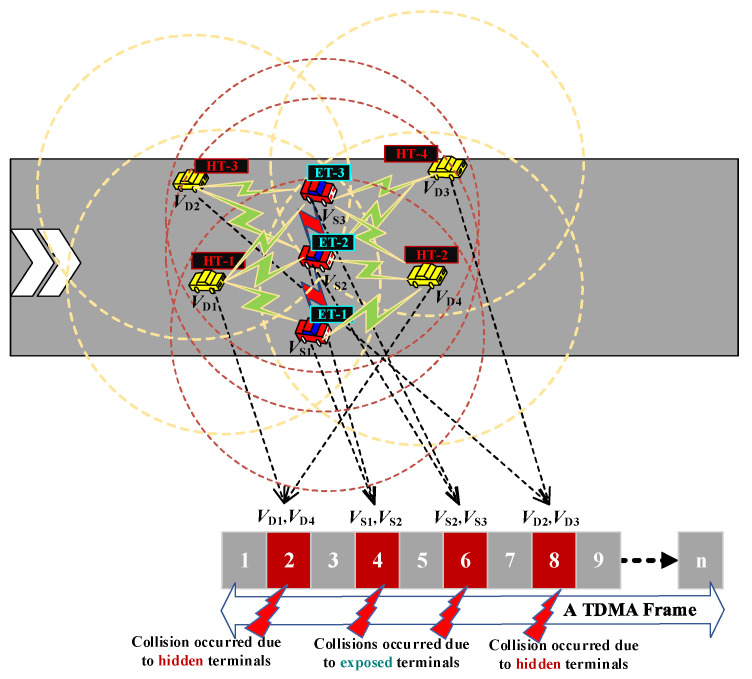
Aggregated collisions of HTs and ETs at associated positions.

**Figure 12 sensors-23-01033-f012:**
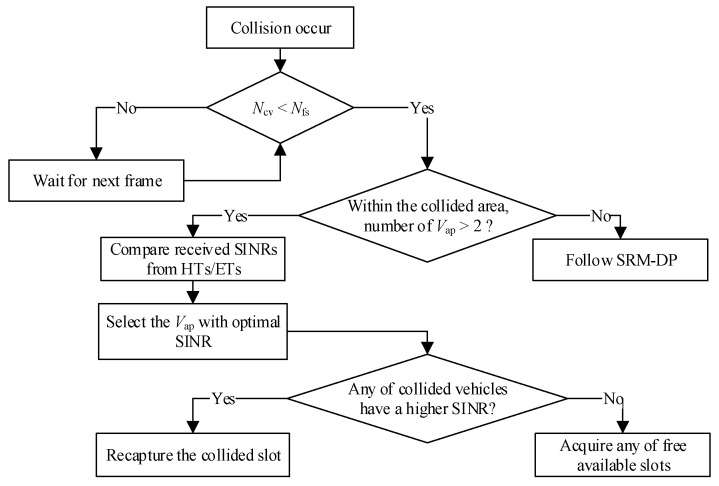
Slot reallocation mechanism for aggregated heterogenous collisions of HTs and ETs at the associated position.

**Figure 13 sensors-23-01033-f013:**
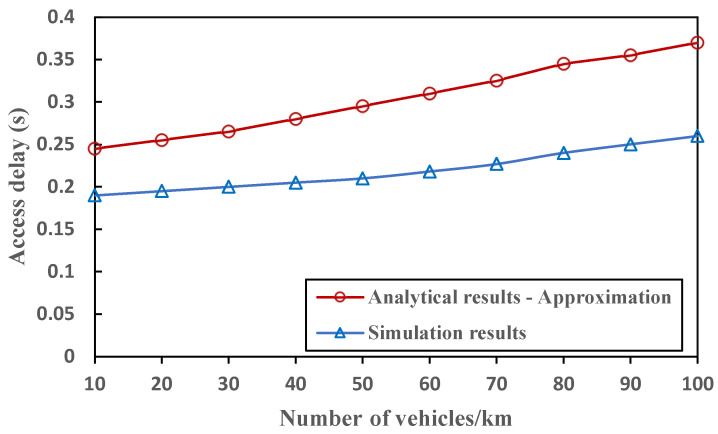
Comparison of access delay for simulation and analytical results.

**Figure 14 sensors-23-01033-f014:**
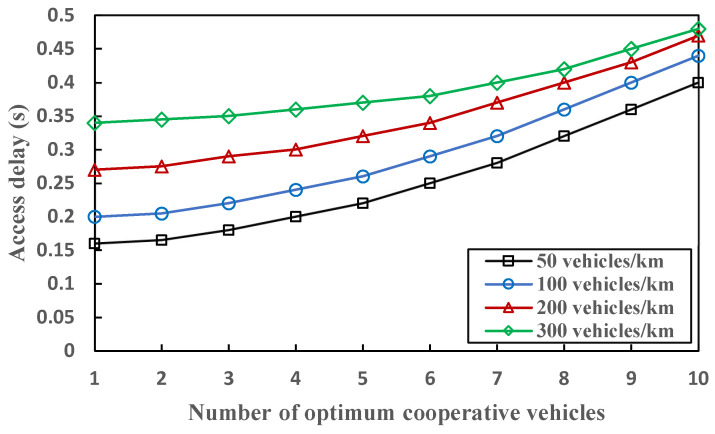
Comparison of access delay with the growing number of vehicle densities related to optimum cooperative vehicles.

**Figure 15 sensors-23-01033-f015:**
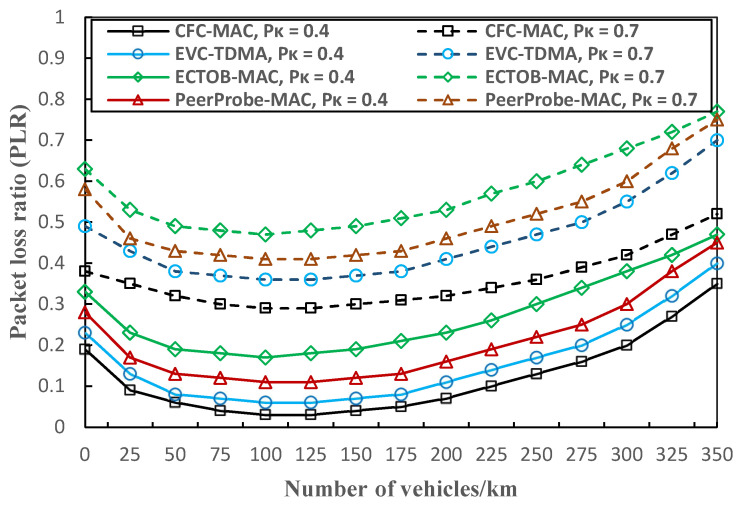
Comparison of packet loss ratio for different protocols.

**Figure 16 sensors-23-01033-f016:**
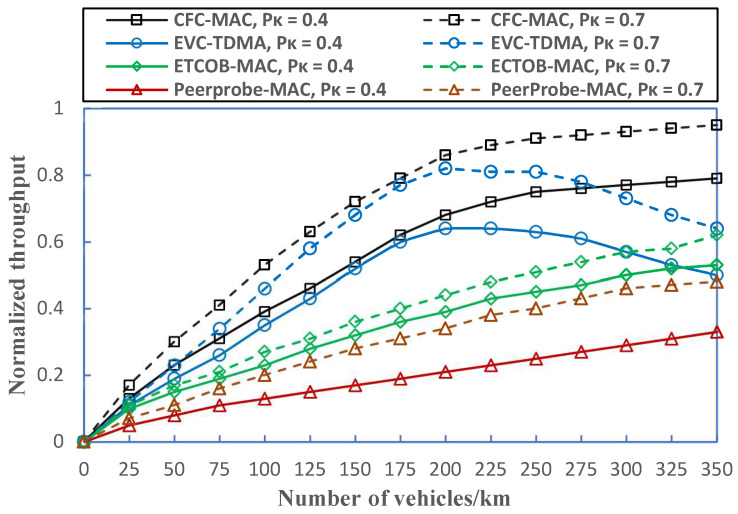
Comparison of throughput for different protocols.

**Figure 17 sensors-23-01033-f017:**
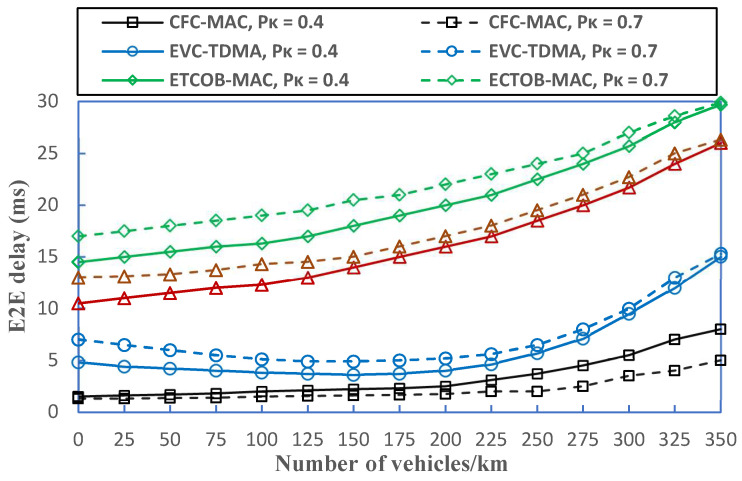
Comparison of E2E delay for different protocols.

**Figure 18 sensors-23-01033-f018:**
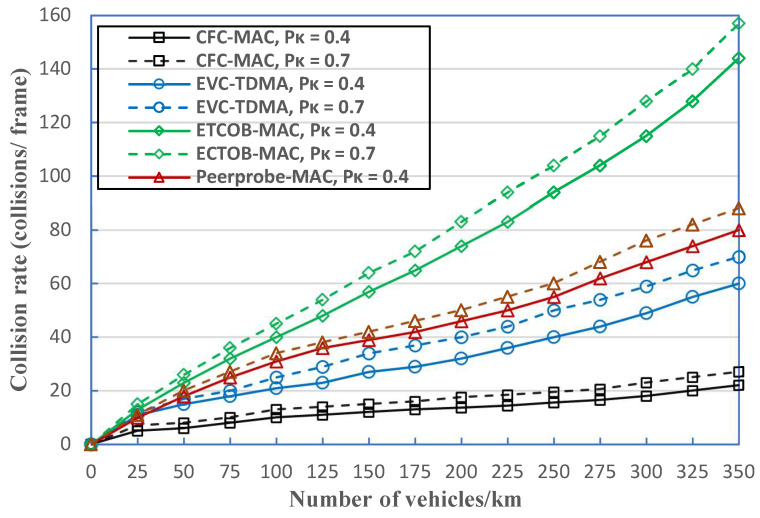
Comparison of collision rates for different protocols.

**Table 1 sensors-23-01033-t001:** Comparison of recent surveys on contention-free cooperative MAC protocols.

Protocol	Beaconing Range	Strategy	Access Delay	Resolving Collisions	Objective
EVC-TDMA [28]	Multi-hop	One-dimensional Markov	Moderate	Merging collisions	Reduce packet dropping rate, improve throughput
NC-MAC [33]	One-hop	Network coding	Low	Hidden terminals, access collisions	Enhance broadcasting reliability
ECTOB-MAC [43]	Multi-hop	Quasi- orthogonal	Moderate	Hidden terminals, access collisions	Improve broadcast performance
TMR-MAC [44]	Multi-hop	Round-trip time	Moderate	Access collisions	Shortest route to destination
MCDP-MAC [45]	Multi-hop	Markov	Moderate	No	High reachability ratio
DARP-MAC [24]	Multi-hop	Adaptive Reservation	Low	Hidden terminals, access collisions	Enhance reliability and scalability
PeerProbe-MAC [46]	Multi-hop	Adaptive compressive sensing	Moderate	No	High-accuracy neighbor distribution estimation

**Table 2 sensors-23-01033-t002:** Vehicle information table of the 1HS, 2HS, and 3HS with adjacent vehicles.

No. of Hop Sets	Vehicle ID	Position	Speed	Moving Direction	Channel No.	Slot No.
1HS/2HS/3HS	[0, 350]	$\left[x_{n},y_{n}\right]$	[20, 120]	Left/Right	[*n-a*, *n-z*]	$\left[0,n_{\tau },2n_{\tau },3n_{\tau }\right]$

**Table 3 sensors-23-01033-t003:** Vehicle parameters in SUMO.

Parameter	Value	Description
MaxSpeed	[80, 200]	The maximum speed that a vehicle can travel (km/h)
Entry 2	[2.2, 18]	The netto-length of a vehicle (m)
Width	[1.6, 2]	The width of a vehicle (m)
Accel	[2, 5]	The acceleration ability of a vehicle (m/s^2^)
Decel	[3, 8]	The deceleration ability of a vehicle (m/s^2^)
MinGap	[2.5, 5]	The gap to the leader when standing in a jam (m)
Sigma	[0.2, 0.7]	The vehicle’s driver imperfection (between 0 and 1)
Tau	[0, 1]	The driver’s desired (minimum) time headway
Car-following model	Karuss	This model describes how a vehicle follows another one
Lane-changing model	LC2013	This model describes how a driver changes lanes

**Table 4 sensors-23-01033-t004:** Transmission power for different vehicle densities.

Duration of a Time Slot	SINR Range	Vehicles/km at *N*_OC_ = 10	Power (dBm) at *N*_OC_ = 10
*L*_b_ = 0.5 ms	[40, 60]	[0.059, 0.375]	[–6, 25]
*L*_b_ = 1 ms	[20, 40]	[0.052, 0.098]	[15.8, 25]
*L*_b_ = 2 ms	[5, 20]	[0.0573, 0.062]	[20.34, 25]
*L*_b_ = 4 ms	[1, 5]	[0.05, 0.0278]	[23.4, 25]

**Table 5 sensors-23-01033-t005:** Simulation parameters.

Parameter	Value	Description
$L_{r}$	5 (km)	Road segment length
$\lambda_{T}$	[0.05, 0.3] (vehicles/m)	Vehicle density
$l_{s}$	4	Lanes/direction
$S_{m}$	27.57 (m/s)	Mean speed value
$R_{b}$	6 (Mbps)	Data rate
$K_{0}$	10^−4.38^	Path loss constant
γ	3.68	Path loss exponent
$P_{t}$	23 (dBm)	Transmission power
$S_{rx}$	−85 (dBm)	Receiver sensitivity
$SINR_{t}$	3.3 (dBm)	Decoding SINR threshold
*W*	180 × 6 (kHz)	Channel bandwidth
$N_{0}$	*W* × 10^−17.4^ (mW)	Noise power
*ρ*	0.5, 0.7, 0.9	Correlation factor
$S_{T}$	500 (s)	Simulation time

## Data Availability

Not applicable.

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
