# Peer review of "A Contention-Free Cooperative MAC Protocol for Eliminating Heterogenous Collisions in Vehicular Ad Hoc Networks"

_sensors, 2023, doi:10.3390/s23021033_

Round 1
Reviewer 1 Report
The topic is interesting, but the study has some technical issues that need to be addressed before it can be published. Some of the important points are given in below:
1. In order to provide a more accurate portrayal of the paper, the entire document should be proofread very carefully, and mistakes should be completely eliminated.
2. The abstract is well-written overall, but it lacks emphasis on the major contributions.
3. The authors should elaborate on the employed methodology with a modified and comprehensive diagram, which should be the paper's primary contribution to predicting critical results.
4. Explain What research gaps is your work intended to fill? What limitations in previous work does it address? Make it in bullet points. What contributions to knowledge does it make to fill the previous gaps?
5. The authors must revise the introduction part of their paper to underline the novelty of their work. It is not clear what is new in the approach used here. What are the previous works that the authors based on their work? What is new? Please use more relevant and recent references with an emphasis on similar works.
6. The Notation of Equations 1, 2, and 3 must be properly cited as the text for the mathematical equations to be properly understood.
7. Authors should include a table containing the vehicle information of the one-hop set (1HS) and two-hop set (2HS) and three-hop sets with adjacent vehicles.
8. What is the relationship between Figure 4 and Equations 3 and 4? Add some explanation to demonstrate the connection between mathematical notation and the figure in the statement.
9. The value column in Table 1 is unclear. How these values are chosen and related to their corresponding parameters.
10. Line 40: Scenario C should be highlighted in contributions.
11. The authors are required to include a comparative graph or table to demonstrate the originality of the concept and the significant contribution it makes.
12. Before the conclusion section, there should be a discussion section.
13. The conclusion section needs to be rewritten so that it takes into account the most important takeaway from all of the topics covered in the paper. Future recommendations must incorporate innovative machine-learning techniques.
Author Response
Response to Reviewer-1 Comments
We greatly thank the reviewer-1 for volunteering your valuable time and providing us with constructive comments. These have helped us to revise and improve the quality of the manuscript. In the following, we respond to the reviewer-1’ comments. We also revise the corresponding part of the manuscript through “Track Changes”.
Comments and Suggestions for Authors
The topic is interesting, but the study has some technical issues that need to be addressed before it can be published. Some of the important points are given in below:
Comment 1: In order to provide a more accurate portrayal of the paper, the entire document should be proofread very carefully, and mistakes should be completely eliminated.
Response 1: Thank you very much for your suggestions. We have consciously proofread the whole manuscript and corrected all the indicated mistakes as your comments. Please refer to the corresponding part of the revised manuscript.
After consciously checking and revising our original manuscript, we have completed some major modifications and also added some state-of-the-arts references including [13], [14], [15], [16], [19], [27], [32], [51], [52] [53] and [55] to the revised manuscript. Moreover, we redrew and modified the Figure 1, Figure 4, Figure 9 and Figure 11 to provide a clearer version. We also made the text inside the Figure 6 to be a bit larger to improve readability. Specifically, we have redrawn the simulation results of Figure 13, Figure 14, Figure 15, Figure 16, Figure 17 and Figure 18 with the clearer and more detailed version.
Comment 2: The abstract is well-written overall, but it lacks emphasis on the major contributions.
Response 2: Thanks for reminding us about this. We have revised the abstract and implemented the major contributions in red as follows:
Abstract: In vehicular ad hoc networks (VANETs), efficient data dissemination to a specified number of vehicles with minimum collisions and limited access delay is critical for accident prevention in road safety. However, packet collisions have a significant impact on access delay and they may lead to unanticipated link failure when a range of diversified collisions are combined due to complex traffic conditions and rapid changes in network topology. In this paper, we propose a distributed contention-free cooperative medium access control (CFC-MAC) protocol to reduce the heterogenous collisions and unintended access delay in stochastic traffic scenarios. Firstly, we develop a cooperative communication system model and cooperative forwarding mechanism to explore the optimum road path between the source and destination by identifying the potential cooperative vehicles. Secondly, we propose a vectorized trajectory estimation mechanism to suppress merging collisions by identifying the relative velocity of the vehicles with different speeds in a specific time interval. Based on the case study, typical heterogeneous collisions and aggregated heterogeneous collisions at dissociated positions and associated positions are investigated. In both cases, we propose the corresponding collision resolving mechanisms by methodically recapturing the collided time slot or acquiring the available free time slots after identifying the access vehicles and comparing the received signal strengths. Performance analysis for collision probability and access delay is given. Finally, simulation results show that the proposed protocol can achieve a deterministic access delay and minimal collision rate, substantially outperforming the existing solutions.
Comment 3: The authors should elaborate on the employed methodology with a modified and comprehensive diagram, which should be the paper's primary contribution to predicting critical results.
Response 3: Thanks for your suggestion, and we find that it does benefit us a lot. Please refer to the corresponding part of the revised manuscript in “Track Changes”. On pp. 8, Lines 313-328, we newly drew a structural flowchart to show the primary contributions of our protocol design and elaborated on it.
Comment 4: Explain What research gaps is your work intended to fill? What limitations in previous work does it address? Make it in bullet points. What contributions to knowledge does it make to fill the previous gaps?
Response 4: Thanks for reminding us about this. Please refer to the corresponding part of the revised manuscript in “Track Changes”.
On pp. 2-3, Lines 94-105, we added some important sentences to describe the two major issues of the research gaps we are planning to fulfill.
And please refer to pp. 5, Lines 212-218, we newly added a table of comparison of recent surveys on contention-free cooperative MAC protocols.
Furthermore, we have revised the contributions of the paper on pp. 3, Lines 120-127. Hopefully, these attempts can satisfy your comment.
Comment 5: The authors must revise the introduction part of their paper to underline the novelty of their work. It is not clear what is new in the approach used here. What are the previous works that the authors based on their work? What is new? Please use more relevant and recent references with an emphasis on similar works.
Response 5: Thanks, we have carefully revised the introduction section to address your comments. Please refer to the corresponding part of the revised manuscript. As mentioned before, on pp. 2, Lines 94-105, we added some important sentences to describe the two major issues of the research gaps we are planning to fulfill. Moreover, we added the most relevant references, please refer to the references [24,26–35] in the revised manuscript. And, more relevant references of [13], [14], [15], [16], [19], [27], [40], [51], [52], [53] and [55] are added to the revised manuscript.
Comment 6: The Notation of Equations 1, 2, and 3 must be properly cited as the text for the mathematical equations to be properly understood.
Response 6: Thank you for your suggestion. We have revised the Notations of Equations 1, 2 and 3, and cited them as the text to be properly understood.
Comment 7: Authors should include a table containing the vehicle information of the one-hop set (1HS) and two-hop set (2HS) and three-hop sets with adjacent vehicles.
Response 7: Thank you for your suggestion. Please refer to the corresponding part of the revised manuscript in “Track Changes”. On pp. 9, Lines 338-342, we provide a table containing the vehicle information of the one-hop set (1HS), two-hop set (2HS) and three-hop sets with adjacent vehicles.
Comment 8: What is the relationship between Figure 4 and Equations 3 and 4? Add some explanation to demonstrate the connection between mathematical notation and the figure in the statement.
Response 8: Thank you for your insightful question. Please refer to the corresponding part of the revised manuscript. We have revised pp. 11 for clarity and added some explanation to prominently describe the connection between mathematical notation and the figure in the statement.
Comment 9: The value column in Table 1 is unclear. How these values are chosen and related to their corresponding parameters.
Response 9: Thanks for your comments. We utilize SUMO to automatically generate simulation scenarios and generate vehicle traffic on the street. The values of the definition of vehicles, vehicle types, and routes listed in Table 1 are chosen according to the SUMO’s website link:
https://sumo.dlr.de/docs/Definition_of_Vehicles%2C_Vehicle_Types%2C_and_Routes.html
Moreover, we utilize a simple, straight street topology in simulations. Vehicles on the street are allowed to change their lanes according to the LC2013 model defined in SUMO, and the car following model is the Karuss model.
Comment 10: Line 40: Scenario C should be highlighted in contributions.
Response 10: Thanks for your valuable suggestion. Please refer to the corresponding part of the revised manuscript. On pp. 3, Lines 125-127, we have highlighted the Scenario C in our contributions.
Comment 11: The authors are required to include a comparative graph or table to demonstrate the originality of the concept and the significant contribution it makes.
Response 11: Thanks for your suggestion, and we find that it does benefit us a lot. Please refer to the corresponding part of the revised manuscript. On pp. 5, Lines 212-218, we added a new table of comparison of the most recent works with surveys on the limitations and challenges. Moreover, on pp. 8, Lines 313-328, we newly drew a structural flowchart to show the primary contributions of our protocol design and elaborated on it.
Comment 12: Before the conclusion section, there should be a discussion section.
Response 12: Thanks for your insightful comment. Please refer to the corresponding part of the revised manuscript. On pp. 31-32, Lines 989-997, we added a discussion section with important sentences.
Comment 13: The conclusion section needs to be rewritten so that it takes into account the most important takeaway from all of the topics covered in the paper. Future recommendations must incorporate innovative machine-learning techniques.
Response 13: We have revised the conclusion according to your suggestion. Please refer to the corresponding part of the revised manuscript in “Track Changes” on pp. 332, Lines 1000-1020.

Reviewer 2 Report
Summary:
This paper introduces a distributed contention-free cooperative medium access control (CFC-MAC) protocol to reduce the heterogenous collisions and unintended access delay in stochastic traffic scenarios.
Strengths:
-the proposed scheme was validated using several experiments.
-the contributions are well defined in the introduction.
-related work section covers all related topics
-the paper is well-written and easy to follow
Weaknesses:
- Some experiment details are missing
comments:
-in the experiment, figure 12-17, I think x-axis unit is #Vehicles/km2 (km square), rather than km
- in section 6.1, it would be good if you elaborate on where did you obtain the map, and what is the nature of the roads, is it Manhattan model or randomly partitioned?-some closely related works are missing
[1] Aung, Nyothiri, et al. "T-Coin: Dynamic traffic congestion pricing system for the Internet of Vehicles in smart cities." Information 11.3 (2020): 149.
[2] Tiwari, Jahnvi, Arun Prakash, and Rajeev Tripathi. "A novel cooperative MAC protocol for safety applications in cognitive radio enabled vehicular ad-hoc networks." Vehicular Communications 29 (2021): 100336.
Author Response
Response to Reviewer-2’ Comments
We greatly thank the reviewer-2 for volunteering your valuable time and providing us with constructive comments. These have helped us to revise and improve the quality of the manuscript. In the following, we respond to the reviewers-2’ comments. We also revise the corresponding part of the manuscript through “Track Changes”.
Comments and Suggestions for Authors
Summary:
This paper introduces a distributed contention-free cooperative medium access control (CFC-MAC) protocol to reduce the heterogenous collisions and unintended access delay in stochastic traffic scenarios.
Strengths:
-the proposed scheme was validated using several experiments.
-the contributions are well defined in the introduction.
-related work section covers all related topics
-the paper is well-written and easy to follow
Weaknesses:
- Some experiment details are missing
Comments:
Comment 1: - In the experiment, figure 12-17, I think x-axis unit is #Vehicles/km2 (km square), rather than km
Response 1: Thank you for your suggestion. The results are from a one-dimension linear network since we are utilizing the SUMO traces to provide microscopic traffic flow simulator by using one-dimensional length with no thickness or width. Therefore, the x-axis unit is in km to simplify the experiment. Your suggestions are well noted and we may attempt to develop our multi-dimension simulations in the future work.
Comment 2: - In section 6.1, it would be good if you elaborate on where did you obtain the map, and what is the nature of the roads, is it Manhattan model or randomly partitioned?-some closely related works are missing
[1] Aung, Nyothiri, et al. "T-Coin: Dynamic traffic congestion pricing system for the Internet of Vehicles in smart cities." Information 11.3 (2020): 149.
[2] Tiwari, Jahnvi, Arun Prakash, and Rajeev Tripathi. "A novel cooperative MAC protocol for safety applications in cognitive radio enabled vehicular ad-hoc networks." Vehicular Communications 29 (2021): 100336.
Response 2: Thank you for your suggestion. We obtained the randomly partitioned map from the OpenStreetMap (OSM) and turned it into a SUMO simulation with INET and VEINs together to simulate a V2V network. And we have added your indicated references to our corresponding paragraphs of the revised manuscript as follows:
- Tiwari, J.; Prakash, A.; Tripathi, R. A Novel Cooperative MAC Protocol for Safety Applications in Cognitive Radio Enabled Vehicular Ad-Hoc Networks. Veh. Commun. 2021, 29, 100336, doi:10.1016/j.vehcom.2021.100336.
- Aung, N.; Zhang, W.; Dhelim, S.; Ai, Y. T-Coin: Dynamic Traffic Congestion Pricing System for the Internet of Vehicles in Smart Cities. Information 2020, 11, 149, doi:10.3390/info11030149.

Reviewer 3 Report
This is a well-written manuscript on Contention-Free Cooperative MAC Protocol for Eliminating Heterogenous Collisions in Vehicular Ad Hoc Networks. It will interest a reasonably broad readership of sensors. I have the following comments that the authors should address before final acceptance:
Point 1: In the introduction section please add motivation and the benefits of your research in bullets.
Point 2:The authors should double-check the spelling and typos in the manuscript.
Point 3:In related work please add the most recent studies on this topic for instance;
F.; Abbasi, R.; Khan, S.; Abid, M.A. An Overview of Medium Access Control and Radio Duty Cycling Protocols for Internet of Things. Electronics 2022, 11, 3873. https://doi.org/10.3390/electronics11233873
In addition, Please add a comparison table and compare at least the most recent studies along with the pros and cons.
Point 4: For all the experiments that the authors did, I think it is necessary to mention how many times the experiments were performed, and add error bars to the figures since the authors are trying to quantify how better their optimizations are.
Point 5: Please add the details of the simulation tools that you used in the experiments.
Point 6:Please add future work in the conclusion section.
Point 7:Please add complete parameters for the equation used in the manuscript.
Author Response
Response to Reviewer-3’ Comments
We greatly thank the reviewer-3 for volunteering their valuable time and providing us with constructive comments. These have helped us to revise and improve the quality of the manuscript. In the following, we respond to the reviewers’ comments. We also revise the corresponding part of the manuscript through “Track Changes”.
Comments and Suggestions for Authors
This is a well-written manuscript on Contention-Free Cooperative MAC Protocol for Eliminating Heterogenous Collisions in Vehicular Ad Hoc Networks. It will interest a reasonably broad readership of sensors. I have the following comments that the authors should address before final acceptance:
Comment 1: In the introduction section please add motivation and the benefits of your research in bullets.
Response 1: Thanks for your suggestion, and we find that it does benefit us a lot. Please refer to the corresponding part of the revised manuscript in “Track Changes”. On pp. 2-3, Lines 94-105, we added some important sentences to describe the two major issues of the research gaps we are planning to fulfill. Moreover, on pp. 3, Lines 120-127, we have revised the contributions of the paper. Hopefully, these attempts can meet the requirements of your comment.
Comment 2: The authors should double-check the spelling and typos in the manuscript.
Response 1: Thank you very much for your suggestions. We have consciously revised the whole manuscript and corrected the spelling and typos as per your comments. Please refer to the corresponding part of the revised manuscript in “Track Changes”.
After consciously checking and revising our original manuscript, we made some major modifications and also added some closely related references including [13], [15], [16], [19], [27], [32], [51], [52] [53] and [54] to the revised manuscript. Moreover, we redrew and modified the Figure 1, Figure 4, Figure 9 and Figure 11 to provide a clearer version. We also made the text inside the Figure 6 to be a bit larger to improve readability. Specifically, we have redrawn the simulation results of Figure 13, Figure 14, Figure 15, Figure 16, Figure 17 and Figure 18 with the clearer and more detailed version.
Comment 3: In related work please add the most recent studies on this topic for instance;
F.; Abbasi, R.; Khan, S.; Abid, M.A. An Overview of Medium Access Control and Radio Duty Cycling Protocols for Internet of Things. Electronics 2022, 11, 3873. https://doi.org/10.3390/electronics11233873
In addition, Please add a comparison table and compare at least the most recent studies along with the pros and cons.
Response 3: Thanks for your suggestion, and we find that it does benefit us a lot. We have added the above reference to our related work as follows:
- Amin, F.; Abbasi, R.; Khan, S.; Abid, M.A. An Overview of Medium Access Control and Radio Duty Cycling Protocols for Internet of Things. Electronics 2022, 11, 3873, doi:10.3390/electronics11233873.
Please refer to the corresponding part of the revised manuscript in “Track Changes”. On pp. 5, Lines 212-218, we added a new table of comparison of the most recent works with surveys with the pros and cons.
Comment 4: For all the experiments that the authors did, I think it is necessary to mention how many times the experiments were performed, and add error bars to the figures since the authors are trying to quantify how better their optimizations are.
Response 4: As we mentioned in our original manuscript, please refer on pp. 20, Lines 770-771, the simulations have been performed ten times, and the standard deviations and average values have been plotted on a graph.
Moreover, we compare the performance metrics of packet loss ratio (PLR), throughput and end-to-end delay between our CFC-MAC and the other three protocols at 0.4 and 0.7 percentages of Pk, which is the probability of the cooperative vehicle receiving k packets. As our simulation results of 350 vehicles/km, about 0.3% and 0.45% errors for PLR at Pk = 0.4 and Pk = 0.7, about 0.15% and 0.03% errors for normalized throughput at Pk = 0.4 and Pk = 0.7, and about 10% and 15% errors for end-to-end delay at Pk = 0.4 and Pk = 0.7, respectively. Therefore, please refer to the corresponding part of the revised manuscript in “Track Changes”. On pp. 31-32, Lines 989-997, we added a discussion section with important sentence as follows:
“CFC-MAC can achieve 5% to 25% performance enhancement thancomparative protocols.”
Comment 5: Please add the details of the simulation tools that you used in the experiments.
Response 5: Thanks for your suggestion, and we find that it does benefit us a lot. We have added the above reference on our related work. Please refer to the corresponding part of the revised manuscript in “Track Changes”. On pp. 23, Lines 793-800, we added the details of the simulation tools that we used in the experiments as follows:
- Omnet++ 5.6.2,
- INET 4.2.8,
- Veins 5.2, and
- SUMO 1.11.0 msi (eclipse)
Comment 6: Please add future work in the conclusion section.
Response 6: We have revised the conclusion according to your suggestion. Please refer to the corresponding part of the revised manuscript in “Track Changes” on pp. 332, Lines 1000-1020.
Comment 7: Please add complete parameters for the equation used in the manuscript.
Response 7: Thank you for reminding us. We have revised and completed the parameters of all the equations used in the manuscript and cited them as the text to be properly understood. Please refer to the corresponding part of the revised manuscript in “Track Changes”

Round 2
Reviewer 3 Report
Thanks, the authors have incorporated all necessary changes.